# Dynamic Model Pruning with Feedback

**Tao Lin**
EPFL, Switzerland
tao.lin@epfl.ch

**Sebastian U. Stich**
EPFL, Switzerland
sebastian.stich@epfl.ch

**Luis Barba**
EPFL & ETH Zurich, Switzerland
luis.barba@inf.ethz.ch

**Daniil Dmitriev**
EPFL, Switzerland
daniil.dmitriev@epfl.ch

**Martin Jaggi**
EPFL, Switzerland
martin.jaggi@epfl.ch

## Abstract

Deep neural networks often have millions of parameters. This can hinder their
deployment to low-end devices, not only due to high memory requirements but also
because of increased latency at inference. We propose a novel model compression
method that generates a sparse trained model without additional overhead: by al-
lowing (i) dynamic allocation of the sparsity pattern and (ii) incorporating feedback
signal to reactivate prematurely pruned weights we obtain a performant sparse
model in one single training pass (retraining is not needed, but can further improve
the performance). We evaluate our method on CIFAR-10 and ImageNet, and show
that the obtained sparse models can reach the state-of-the-art performance of dense
models. Moreover, their performance surpasses that of models generated by all
previously proposed pruning schemes.

## 1 Introduction

Highly overparametrized deep neural networks show impressive results on machine learning tasks.
However, with the increase in model size comes also the demand for memory and computer power at
inference stage—two resources that are scarcely available on low-end devices. Pruning techniques
have been successfully applied to remove a significant fraction of the network weights while preserv-
ing test accuracy attained by dense models. In some cases, the generalization of compressed networks
has even been found to be better than with full models (Han et al., 2015; 2017; Mocanu et al., 2018).

The *sparsity* of a network is the number of weights that are identically zero, and can be obtained
by applying a *sparsity mask* on the weights. There are several different approaches to find sparse
models. For instance, *one-shot pruning* strategies find a suitable sparsity mask by inspecting the
weights of a pretrained network (Mozer & Smolensky, 1989; LeCun et al., 1990; Han et al., 2017).
While these algorithms achieve a substantial size reduction of the network with little degradation
in accuracy, they are computationally expensive (training and refinement on the dense model), and
they are outperformed by algorithms that explore different sparsity masks instead of a single one.
In *dynamic pruning* methods, the sparsity mask is readjusted during training according to different
criteria (Mostafa & Wang, 2019; Mocanu et al., 2018). However, these methods require fine-tuning
of many hyperparameters.

We propose a new pruning approach to obtain sparse neural networks with state-of-the-art test
accuracy. Our compression scheme uses a new saliency criterion that identifies important weights in
the network throughout training to propose candidate masks. As a key feature, our algorithm not only
evolves the pruned sparse model alone, but jointly also a (closely related) dense model that is used
in a natural way to correct for pruning errors during training. This results in better generalization
properties on a wide variety of tasks, since the simplicity of the scheme allows us further to study it
from a theoretical point of view, and to provide further insights and interpretation. We do not require

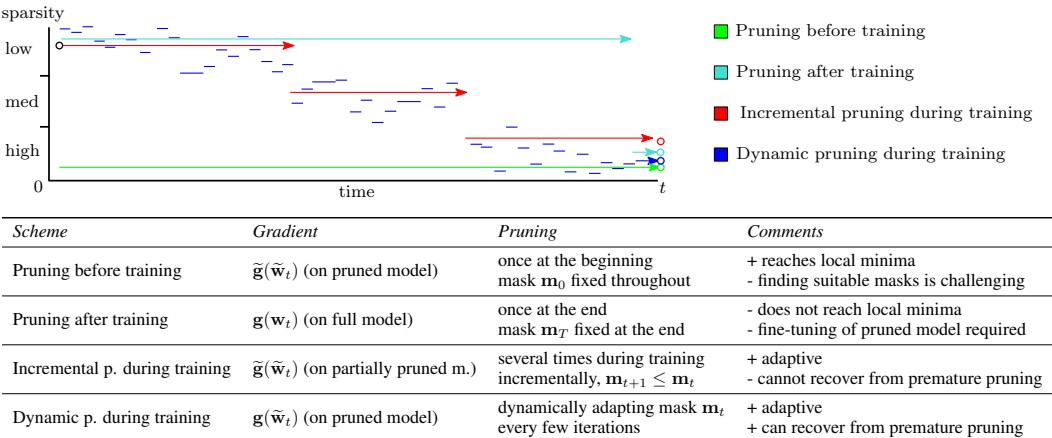

| Scheme | Gradient | Pruning | Comments |
|---|---|---|---|
| Pruning before training | $\widetilde{\mathbf{g}}(\widetilde{\mathbf{w}}_t)$ (on pruned model) | once at the beginning mask $\mathbf{m}_0$ fixed throughout | + reaches local minima - finding suitable masks is challenging |
| Pruning after training | $\mathbf{g}(\mathbf{w}_t)$ (on full model) | once at the end mask $\mathbf{m}_T$ fixed at the end | - does not reach local minima - fine-tuning of pruned model required |
| Incremental p. during training | $\widetilde{\mathbf{g}}(\widetilde{\mathbf{w}}_t)$ (on partially pruned m.) | several times during training incrementally, $\mathbf{m}_{t+1} \leq \mathbf{m}_t$ | + adaptive - cannot recover from premature pruning |
| Dynamic p. during training | $\mathbf{g}(\widetilde{\mathbf{w}}_t)$ (on pruned model) | dynamically adapting mask $\mathbf{m}_t$ every few iterations | + adaptive + can recover from premature pruning |

Figure 1: Schematic view of different pruning methodologies and their properties.

tuning of additional hyperparameters, and no retraining of the sparse model is needed (though can further improve performance).

**Contributions.**

- A novel dynamic pruning scheme, that incorporates an error feedback in a natural way     Sec. 3
  and finds a trained sparse model in one training pass.     Sec. 5
- We demonstrate state-of-the-art performance (in accuracy and sparsity),     Sec. 5
  ourperforming all previously proposed pruning schemes.     Sec. 5
- We complement our results by an ablation study that provides further insights.     Sec. 6
  and convergence analysis for convex and non-convex objectives.     Sec. 4

## 2   RELATED WORK

Previous works on obtaining pruned networks can (loosely) be divided into three main categories.

**Pruning after training.** Training approaches to obtain sparse networks usually include a three stage pipeline—training of a dense model, *one-shot pruning* and fine-tuning—e.g., (Han et al., 2015). Their results (i.e., moderate sparsity level with minor quality loss) made them the standard method for network pruning and led to several variations (Guo et al., 2016; Carreira-Perpinán & Idelbayev, 2018).

**Pruning during training.** Zhu & Gupta (2017) propose the use of magnitude-based pruning and to gradually increase the sparsity ratio while training the model from scratch. A pruning schedule determines when the new masks are computed (extending and simplifying (Narang et al., 2017)). He et al. (2018) (SFP) prune entire filters of the model at the end of each epoch, but allow the pruned filters to be updated when training the model. Deep Rewiring (DeepR) (Bellec et al., 2018) allows for even more adaptivity by performing pruning and regrowth decisions periodically. This approach is computationally expensive and challenging to apply to large networks and datasets. Sparse evolutionary training (SET) (Mocanu et al., 2018) simplifies prune–regrowth cycles by using heuristics for random growth at the end of each training epoch and NeST (Dai et al., 2019) by inspecting gradient magnitudes.

Dynamic Sparse Reparameterization (DSR) (Mostafa & Wang, 2019) implements a prune–redistribute–regrowth cycle where target sparsity levels are redistributed among layers, based on loss gradients (in contrast to SET, which uses fixed, manually configured, sparsity levels). Sparse Momentum (SM) (Dettmers & Zettlemoyer, 2019) follows the same cycle but instead using the mean momentum magnitude of each layer during the redistribute phase. SM outperforms DSR on ImageNet for unstructured pruning by a small margin but has no performance difference on CIFAR experiments. Our approach also falls in the dynamic category but we use error compensation mechanisms instead of hand crafted redistribute–regrowth cycles.

**Pruning before training.** Recently—spurred by the lottery ticket hypothesis (LT) (Frankle & Carbin, 2019)—methods which try to find a sparse mask that can be trained from scratch have attracted increased interest. For instance, Lee et al. (2019) propose SNIP to find a pruning mask by inspecting connection sensitivities and identifying structurally important connections in the network for a given task. Pruning is applied at initialization, and the sparsity mask remains fixed throughout training.

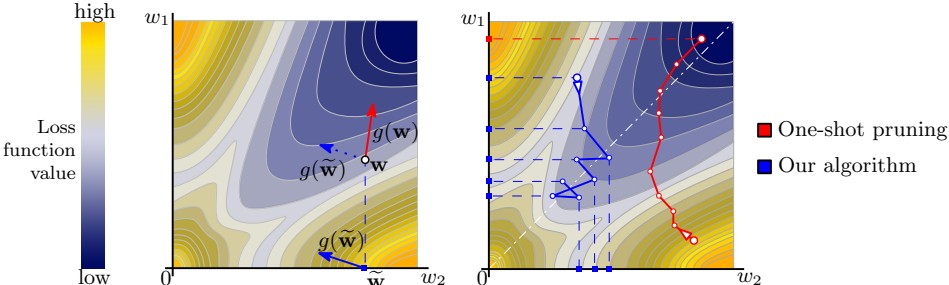

Figure 2: **Left**: One-shot pruning (red) computes a stochastic gradient at $\mathbf{w}$ and takes a step towards the best dense model. In contrast, DPF (blue) computes a stochastic gradient at the pruned model $\widetilde{\mathbf{w}}$ (here obtained by smallest magnitude pruning), and takes a step that best suits the compressed model. **Right**: One-shot pruning commits to a single sparsity mask and might obtain sparse models that generalize poorly (without retraining). DPF explores different available sparsity patterns and finds better sparse models.

Note that Frankle & Carbin (2019); Frankle et al. (2019) do not propose an efficient pruning scheme to find the mask, instead they rely on iterative pruning, repeated for several full training passes.

**Further Approaches.** Srinivas et al. (2017); Louizos et al. (2018) learn gating variables (e.g. through $\ell_0$ regularization) that minimize the number of nonzero weights, recent parallel work studies filter pruning for pre-trained models (You et al., 2019). Gal et al. (2017); Neklyudov et al. (2017); Molchanov et al. (2017) prune from Bayesian perspectives to learn dropout probabilities during training to prune and sparsify networks as dropout weight probabilities reach 1. Gale et al. (2019) extensively study recent unstructured pruning methods on large-scale learning tasks, and find that complex techniques (Molchanov et al., 2017; Louizos et al., 2018) perform inconsistently. Simple magnitude pruning approaches achieve comparable or better results (Zhu & Gupta, 2017).

## 3 METHOD

We consider the training of a non-convex loss function $f\colon \mathbb{R}^d \to \mathbb{R}$. We assume for a weight vector $\mathbf{w} \in \mathbb{R}^d$ to have access to a stochastic gradient $\mathbf{g}(\mathbf{w}) \in \mathbb{R}^d$ such that $\mathbb{E}[\mathbf{g}(\mathbf{w})] = \nabla f(\mathbf{w})$. This corresponds to the standard machine learning setting with $\mathbf{g}(\mathbf{w})$ representing a (mini-batch) gradient of one (or several) components of the loss function. Stochastic Gradient Descent (SGD) computes a sequence of iterates by the update rule

$$\mathbf{w}_{t+1} := \mathbf{w}_t - \gamma_t \mathbf{g}(\mathbf{w}_t)\,, \tag{SGD}$$

for some learning rate $\gamma_t$. To obtain a sparse model, a general approach is to *prune* some of the weights of $\mathbf{w}_t$, i.e., to set them to zero. Such pruning can be implemented by applying a *mask* $\mathbf{m} \in \{0,1\}^d$ to the weights, resulting in a sparse model $\widetilde{\mathbf{w}}_t := \mathbf{m} \odot \mathbf{w}_t$, where $\odot$ denotes the entry-wise (Hadamard) product. The mask could potentially depend on the weights $\mathbf{w}_t$ (e.g., smallest magnitude pruning), or depend on $t$ (e.g., the sparsity is incremented over time).

Before we introduce our proposed dynamic pruning scheme, we formalize the three main existing types of pruning methodologies (summarized in Figure 1). These approaches differ in the way the mask is computed, and the moment when it is applied.[1]

**Pruning before training.** A mask $\mathbf{m}_0$ (depending on e.g. the initialization $\mathbf{w}_0$ or the network architecture of $f$) is applied and (SGD) is used for training on the resulting subnetwork $\widetilde{f}(\mathbf{w}) := f(\mathbf{m}_0 \odot \mathbf{w})$ with the advantage that only pruned weights need to be stored and updated[2], and that by training with SGD a local minimum of the subnetwork $\widetilde{f}$ (but not of $f$—the original training target) can be reached. In practice however, it remains a challenge to efficiently determine a good mask $\mathbf{m}_0$ and a wrongly chosen mask at the beginning strongly impacts the performance.

---

[1] The method introduced in Section 2 typically follow one of these broad themes loosely, with slight variations in detail. For the sake of clarity we omit a too technical and detailed discussion here.

[2] When training on $\widetilde{f}(\mathbf{w})$, it suffices to access stochastic gradients of $\widetilde{f}(\mathbf{w})$, denoted by $\widetilde{\mathbf{g}}(\mathbf{w})$, which can potentially be cheaper be computed than by naively applying the mask to $\mathbf{g}(\mathbf{w})$ (note $\widetilde{\mathbf{g}}(\mathbf{w}) = \mathbf{m}_0 \odot \mathbf{g}(\mathbf{w})$).

**Pruning after training (one-shot pruning).** A dense model is trained, and pruning is applied to the trained model $\mathbf{w}_T$. As the pruned model $\widetilde{\mathbf{w}}_T = \mathbf{m}_T \odot \mathbf{w}_T$ is very likely not at a local optimum of $f$, fine-tuning (retraining with the fixed mask $\mathbf{m}_T$) is necessary to improve performance.

**Pruning during training (incremental and dynamic pruning).** Dynamic schemes change the mask $\mathbf{m}_t$ every (few) iterations based on observations during training (i.e. by observing the weights and stochastic gradients). Incremental schemes monotonically increase the sparsity pattern, fully dynamic schemes can also reactivate previously pruned weights. In contrast to previous dynamic schemes that relied on elaborated heuristics to adapt the mask $\mathbf{m}_t$, we propose a simpler approach:

**Dynamic pruning with feedback (DPF, Algorithm 1).** Our scheme evaluates a stochastic gradient at the *pruned* model $\widetilde{\mathbf{w}}_t = \mathbf{m}_t \odot \mathbf{w}_t$ and applies it to the (simultaneously maintained) dense model $\mathbf{w}_t$:

$$\mathbf{w}_{t+1} := \mathbf{w}_t - \gamma_t \mathbf{g}(\mathbf{m}_t \odot \mathbf{w}_t) = \mathbf{w}_t - \gamma_t \mathbf{g}(\widetilde{\mathbf{w}}_t) \,. \tag{DPF}$$

Applying the gradient to the full model allows to recover from "errors", i.e. prematurely masking out important weights: when the accumulated gradient updates from the following steps drastically change a specific weight, it can become activated again (in contrast to incremental pruning approaches that have to stick to sub-optimal decisions). For illustration, observe that (DPF) can equivalently be written as

$$\mathbf{w}_{t+1} = \mathbf{w}_t - \gamma_t \mathbf{g}(\mathbf{w}_t + \mathbf{e}_t),$$

where $\mathbf{e}_t := \widetilde{\mathbf{w}}_t - \mathbf{w}_t$ is the *error* produced by the compression. This provides a different intuition of the behavior of (DPF), and connects it with the concept of *error-feedback* (Stich et al., 2018; Karimireddy et al., 2019).[3] We illustrate this principle in Figure 2 and give detailed pseudocode and further implementation details in Appendix A.1. The DPF scheme can also be seen as an instance of a more general class of schemes that apply (arbitrary) perturbed gradient updates to the dense model. For instance straight-through gradient estimators (Bengio et al., 2013) that are used to empirically simplify the backpropagation can be seen as such perturbations. Our stronger assumptions on the structure of the perturbation allow to derive non-asymptotic convergence rates in the next section, though our analysis could also be extended to the setting in (Yin et al., 2019) if the perturbations can be bounded.

## 4  CONVERGENCE ANALYSIS

We now present convergence guarantees for (DPF). For the purposes of deriving theoretical guarantees, we assume that the training objective is smooth, that is $\|\nabla f(\mathbf{w}) - \nabla f(\mathbf{v})\| \le L \|\mathbf{w} - \mathbf{v}\|$, $\forall \mathbf{w}, \mathbf{v} \in \mathbb{R}^d$, for a constant $L > 0$, and that the stochastic gradients are bounded $\mathbb{E} \|\mathbf{g}(\widetilde{\mathbf{w}}_t)\|^2 \le G^2$ for every pruned model $\widetilde{\mathbf{w}}_t = \mathbf{m}_t(\mathbf{w}_t) \odot \mathbf{w}_t$. The *quality* of this pruning is defined as the parameter $\delta_t \in [0, 1]$ such that

$$\delta_t := \|\mathbf{w}_t - \widetilde{\mathbf{w}}_t\|^2 \big/ \|\mathbf{w}_t\|^2 \,. \tag{1}$$

Pruning without information loss corresponds to $\widetilde{\mathbf{w}}_t = \mathbf{w}_t$, i.e., $\delta_t = 0$, and in general $\delta_t \le 1$.

**Convergence on Convex functions.** We first consider the case when $f$ is in addition $\mu$-strongly convex, that is $\langle \nabla f(\mathbf{v}), \mathbf{w} - \mathbf{v} \rangle \le f(\mathbf{w}) - f(\mathbf{v}) - \frac{\mu}{2} \|\mathbf{w} - \mathbf{v}\|^2$, $\forall \mathbf{w}, \mathbf{v} \in \mathbb{R}^d$. While it is clear that this assumption does not apply to neural networks, it eases the presentation as strongly convex functions have a unique (global) minimizer $\mathbf{w}^\star := \arg\min_{\mathbf{w} \in \mathbb{R}^d} f(\mathbf{w})$.

**Theorem 4.1.** *Let $f$ be $\mu$-strongly convex and learning rates given as $\gamma_t = \frac{4}{\mu(t+2)}$. Then for a randomly chosen pruned model $\widetilde{\mathbf{u}}$ of the iterates $\{\widetilde{\mathbf{w}}_0, \dots, \widetilde{\mathbf{w}}_T\}$ of DPF, concretely $\widetilde{\mathbf{u}} = \widetilde{\mathbf{w}}_t$ with probability $p_t = \frac{2(t+1)}{(T+1)(T+2)}$, it holds that—in expectation over the stochasticity and the selection of $\widetilde{\mathbf{u}}$:*

$$\mathbb{E} f(\widetilde{\mathbf{u}}) - f(\mathbf{w}^\star) = \mathcal{O}\left(\frac{G^2}{\mu T} + L\mathbb{E}\left[\delta_t \|\mathbf{w}_t\|^2\right]\right) \,. \tag{2}$$

The rightmost term in (2) measures the average quality of the pruning. However, unless $\delta_t \to 0$ or $\|\mathbf{w}_t\| \to 0$ for $t \to \infty$, the error term never completely vanishes, meaning that the method converges only to a neighborhood of the optimal solution (this not only holds for the pruned model, but also for

---

[3]Our variable $\mathbf{w}_t$ corresponds to $\tilde{\mathbf{x}}_t$ in the notation of Karimireddy et al. (2019). Their error-fixed SGD algorithm evaluates gradients at perturbed iterates $\mathbf{x}_t := \tilde{\mathbf{x}}_t + \mathbf{e}_t$, which correspond precisely to $\tilde{\mathbf{w}}_t = \mathbf{w}_t + \mathbf{e}_t$ in our notation. This shows the connection of these two methods.

the jointly maintained dense model, as we will show in the appendix). This behavior is expected, as the global optimal model $\mathbf{w}^\star$ might be dense and cannot be approximated well by a sparse model.

For one-shot methods that only prune the final (SGD) iterate $\mathbf{w}_T$ at the end, we have instead:

$$\mathbb{E}f(\widetilde{\mathbf{w}}_t) - f(\mathbf{w}^\star) \leq 2\mathbb{E}\left(f(\mathbf{w}_T) - f(\mathbf{w}^\star)\right) + L\delta_T \mathbb{E}\left\|\mathbf{w}_T\right\|^2 = \mathcal{O}\left(\frac{LG^2}{\mu^2 T} + L\mathbb{E}\left[\delta_T \left\|\mathbf{w}_T\right\|^2\right]\right),$$

as we show in the appendix. First, we see from this expression that the estimate is very sensitive to $\delta_T$ and $\mathbf{w}_T$, i.e. the quality of the pruning the final model. This could be better or worse than the average of the pruning quality of all iterates. Moreover, one looses also a factor of the condition number $\frac{L}{\mu}$ in the asymptotically decreasing term, compared to (2). This is due to the fact that standard convergence analysis only achieves optimal rates for an average of the iterates (but not the last one). This shows a slight theoretical advantage of DPF over rounding at the end.

**Convergence on Non-Convex Functions to Stationary Points.** Secondly, we consider the case when $f$ is a non-convex function and show convergence (to a neighborhood) of a stationary point.

**Theorem 4.2.** *Let learning rate be given as* $\gamma = \frac{c}{\sqrt{T}}$, *for* $c = \sqrt{\frac{f(\mathbf{w}_0) - f(\mathbf{w}^\star)}{LG^2}}$. *Then for pruned model* $\widetilde{\mathbf{u}}$ *chosen uniformly at random from the iterates* $\{\widetilde{\mathbf{w}}_0, \ldots, \widetilde{\mathbf{w}}_T\}$ *of* DPF, *concretely* $\widetilde{\mathbf{u}} := \widetilde{\mathbf{w}}_t$ *with probability* $p_t = \frac{1}{T+1}$, *it holds—in expectation over the stochasticity and the selection of* $\widetilde{\mathbf{u}}$:

$$\mathbb{E}\left\|\nabla f(\widetilde{\mathbf{u}})\right\|^2 = \mathcal{O}\left(\frac{\sqrt{L(f(\mathbf{w}_0) - f(\mathbf{w}^\star))}G}{\sqrt{T}} + L^2 \mathbb{E}\left[\delta_t \left\|\mathbf{w}_t\right\|^2\right]\right). \tag{3}$$

**Extension to Other Compression Schemes.** So far we put our focus on simple mask pruning schemes to achieve high model sparsity. However, the pruning scheme in Algorithm 1 could be replaced by an arbitrary compressor $\mathcal{C}\colon \mathbb{R}^d \to \mathbb{R}^d$, i.e., $\widetilde{\mathbf{w}}_t = \mathcal{C}(\mathbf{w}_t)$. Our analysis extends to compressors as e.g. defined in (Karimireddy et al., 2019; Stich & Karimireddy, 2019), whose quality is also measured in terms of the $\delta_t$ parameters as in (1). For example, if our objective was not to obtain a sparse model, but to produce a quantized neural network where inference could be computed faster on low-precision numbers, then we could define $\mathcal{C}$ as a quantized compressor. One variant of this approach is implemented in the Binary Connect algorithm (BC) (Courbariaux et al., 2015) with prominent results, see also (Li et al., 2017) for further insights and discussion.

## 5 EXPERIMENTS

We evaluated DPF together with its competitors on a wide range of neural architectures and sparsity levels. **DPF exhibits consistent and noticeable performance benefits over its competitors.**

### 5.1 EXPERIMENTAL SETUP

**Datasets.** We evaluated DPF on two image classification benchmarks: (1) CIFAR-10 (Krizhevsky & Hinton, 2009) (50K/10K training/test samples with 10 classes), and (2) ImageNet (Russakovsky et al., 2015) (1.28M/50K training/validation samples with 1000 classes). We adopted the standard data augmentation and preprocessing scheme from He et al. (2016a); Huang et al. (2016); for further details refer to Appendix A.2.

**Models.** Following the common experimental setting in related work on network pruning (Liu et al., 2019; Gale et al., 2019; Dettmers & Zettlemoyer, 2019; Mostafa & Wang, 2019), our main experiments focus on ResNet (He et al., 2016a) and WideResNet (Zagoruyko & Komodakis, 2016). However, DPF can be effectively extended to other neural architectures, e.g., VGG (Simonyan & Zisserman, 2015), DenseNet (Huang et al., 2017). We followed the common definition in He et al. (2016a); Zagoruyko & Komodakis (2016) and used ResNet-$a$, WideResNet-$a$-$b$ to represent neural network with $a$ layers and width factor $b$.

**Baselines.** We considered the state-of-the-art model compression methods presented in the table below as our strongest competitors. We omit the comparison to other dynamic reparameterization methods, as DSR can outperform DeepR (Bellec et al., 2018) and SET (Mocanu et al., 2018) by a noticeable margin (Mostafa & Wang, 2019).

**Implementation of DPF.** Compared to other dynamic reparameterization methods (e.g. DSR and SM) that introduced many extra hyperparameters, our method has trivial hyperparameter tuning

| Scheme | Reference | Pruning | How the mask(s) are found |
|---|---|---|---|
| Lottery Ticket (LT) 
 SNIP | 2019, FDRC 
 2019, LAT | before training | 10-30 successive rounds of (full training + pruning). 
 By inspecting properties/sensitivity of the network. |
| One-shot + fine-tuning (One-shot P+FT) | 2015, HPDT | after training | Saliency criterion (prunes smallest weights). |
| Incremental pruning + fine-tuning (Incremental) | 2017, ZG | incremental | Saliency criterion. Sparsity is gradually incremented. |
| Dynamic Sparse Reparameterization (DSR) 
 Sparse Momentum (SM) 
 DPF | 2019, MW 
 2019, DZ 
 *ours* | dynamic | Prune–redistribute–regrowth cycle. 
 Prune–redistribute–regrowth + mean momentum. 
 Reparameterization via error feedback. |

overhead. We perform pruning across all neural network layers (no layer-wise pruning) using magnitude-based unstructured weight pruning (inherited from Han et al. (2015)). We found the best preformance when updating the mask every 16 iterations (see also Table 11) and we keep this value fixed for all experiments (independent of the architecture or task).

Unlike our competitors that may ignore some layers (e.g. the first convolution and downsampling layers in DSR), we applied DPF (as well as the One-shot P+FT and Incremental baselines) to all convolutional layers while keeping the last fully-connected layer[4], biases and batch normalization layers dense. Lastly, our algorithm gradually increases the sparsity $s_t$ of the mask from 0 to the desired sparsity using the same scheduling as in (Zhu & Gupta, 2017); see Appendix A.2.

**Training schedules.** For all competitors, we adapted their open-sourced code and applied a consistent (and standard) training scheme over different methods to ensure a fair comparison. Following the standard training setup for CIFAR-10, we trained ResNet-$a$ for 300 epochs and decayed the learning rate by 10 when accessing 50% and 75% of the total training samples (He et al., 2016a; Huang et al., 2017); and we trained WideResNet-$a$-$b$ as Zagoruyko & Komodakis (2016) for 200 epochs and decayed the learning rate by 5 when accessing 30%, 60% and 80% of the total training samples. For ImageNet training, we used the training scheme in (Goyal et al., 2017) for 90 epochs and decayed learning rate by 10 at 30, 60, 80 epochs. For all datasets and models, we used mini-batch SGD with Nesterov momentum (factor 0.9) with fine-tuned learning rate for DPF. We reused the tuned (or recommended) hyperparameters for our baselines (DSR and SM), and fine-tuned the optimizer and learning rate for One-shot P+FT, Incremental and SNIP. The mini-batch size is fixed to 128 for CIFAR-10 and 1024 for ImageNet regardless of datasets, models and methods.

## 5.2 EXPERIMENT RESULTS

**CIFAR-10.** Figure 3 shows a comparison of different methods for WideResNet-28-2. For low sparsity level (e.g. 50%), DPF outperforms even the dense baseline, which is in line with regularization properties of network pruning. Furthermore, DPF can prune the model up to a very high level (e.g. 99%), and still exhibit viable performance. This observation is also present in Table 1, where the results of training different state-of-the-art DNN architectures with higher sparsities are depicted. DPF shows reasonable performance even with extremely high sparsity level on large models (e.g. WideResNet-28-8 with sparsity ratio 99.9%), while other methods either suffer from significant quality loss or even fail to converge.

Because simple model pruning techniques sometimes show better performance than complex techniques (Gale et al., 2019), we further consider these simple models in Table 2. While DPF outperforms them in almost all settings, it faces difficulties pruning smaller models to extremely high sparsity ratios (e.g. ResNet-20 with sparsity ratio 95%).This seems however to be an artifact of fine-tuning, as DPF with extra fine-tuning convincingly outperforms all other methods regardless of the network size. This comes to no surprise as schemes like One-shot P+FT and Incremental do not benefit from extra fine-tuning, since it is already incorporated in their training procedure and they might become stuck in local minima. On the other hand, dynamic pruning methods, and in particular DPF, work on a different paradigm, and can still heavily benefit from fine-tuning.[5]

Figure 13 (in Appendix A.3.4) depicts another interesting property of DPF. When we search for a subnetwork with a (small) predefined number of parameters for a fixed task, it is much better to run

---

[4]The last fully-connected layer normally makes up only a very small faction of the total MACs, e.g. 0.05% for ResNet-50 on ImageNet and 0.0006% for WideResNet-28-2 on CIFAR-10.

[5]Besides that a large fraction of the mask elements already converge during training (see e.g. Figure 4 below), not all mask elements converge. Thus DPF can still benefit from fine-tuning on the fixed sparsity mask.

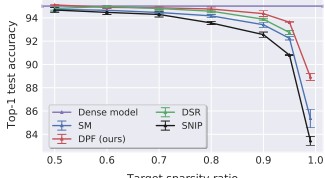 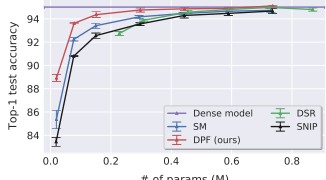 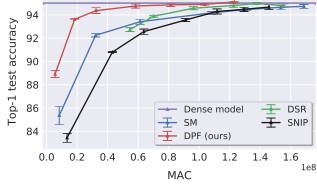

(a) Sparsity ratio v.s. top-1 test acc.    (b) # of params v.s. top-1 test acc.    (c) MAC v.s. top-1 test acc.

Figure 3: Top-1 test accuracy of **WideResNet-28-2** on **CIFAR-10** for unstructured weight pruning. The original model has 1.47M parameters with 216M MACs (Multiplier-ACcumulator). We varied the sparsity ratio from 50% to 99%. The complete numerical test accuracy values refer to Table 4 in Appendix A.3.1. The lower # of params and MACs the model has, the higher sparsity ratio it uses. All results are averaged over three runs. Note that different methods might consider different types of layers and thus the same pruning sparsity ratio might result in the slight difference for both of # of params and MACs. The DSR cannot converge when using the extreme high sparsity ratio (99%).

Table 1: Top-1 test accuracy of SOTA DNNs on **CIFAR-10** for unstructured weight pruning. We considered unstructured pruning and the $\star$ indicates the method cannot converge. The results are averaged for three runs. The results we presented for each model consider some reasonable pruning ratios (we prune more aggressively for deeper and wider neural networks), and readers can refer to Table 4 (in Appendix A.3.1) for a complete overview.

| Model | Baseline on dense model | Methods | | | | Target Pr. ratio |
|---|---|---|---|---|---|---|
| | | SNIP (L$^+$, 2019) | SM (DZ, 2019) | DSR (MW, 2019) | DPF | |
| VGG16-D | $93.74 \pm 0.13$ | $93.04 \pm 0.26$ | $93.59 \pm 0.17$ | - | $\mathbf{93.87} \pm 0.15$ | 95% |
| ResNet-20 | $92.48 \pm 0.20$ | $91.10 \pm 0.22$ | $91.98 \pm 0.01$ | $92.00 \pm 0.19$ | $\mathbf{92.42} \pm 0.14$ | 70% |
| | | $90.53 \pm 0.27$ | $91.54 \pm 0.16$ | $91.78 \pm 0.28$ | $\mathbf{92.17} \pm 0.21$ | 80% |
| | | $88.50 \pm 0.13$ | $89.76 \pm 0.40$ | $87.88 \pm 0.04$ | $\mathbf{90.88} \pm 0.07$ | 90% |
| | | $84.91 \pm 0.25$ | $83.03 \pm 0.74$ | $\star$ | $\mathbf{88.01} \pm 0.30$ | 95% |
| ResNet-32 | $93.83 \pm 0.12$ | $90.40 \pm 0.26$ | $91.54 \pm 0.18$ | $91.41 \pm 0.23$ | $\mathbf{92.42} \pm 0.18$ | 90% |
| | | $87.23 \pm 0.29$ | $88.68 \pm 0.22$ | $84.12 \pm 0.32$ | $\mathbf{90.94} \pm 0.35$ | 95% |
| ResNet-56 | $94.51 \pm 0.20$ | $91.43 \pm 0.34$ | $92.73 \pm 0.21$ | $93.78 \pm 0.20$ | $\mathbf{93.95} \pm 0.11$ | 90% |
| | | $\star$ | $90.96 \pm 0.40$ | $92.57 \pm 0.09$ | $\mathbf{92.74} \pm 0.08$ | 95% |
| WideResNet-28-2 | $95.01 \pm 0.04$ | $92.58 \pm 0.22$ | $93.41 \pm 0.22$ | $93.88 \pm 0.08$ | $\mathbf{94.36} \pm 0.24$ | 90% |
| | | $90.80 \pm 0.04$ | $92.24 \pm 0.14$ | $92.74 \pm 0.17$ | $\mathbf{93.62} \pm 0.05$ | 95% |
| | | $83.45 \pm 0.38$ | $85.36 \pm 0.80$ | $\star$ | $\mathbf{88.92} \pm 0.29$ | 99% |
| WideResNet-28-4 | $95.69 \pm 0.10$ | $93.62 \pm 0.17$ | $94.45 \pm 0.14$ | $94.63 \pm 0.08$ | $\mathbf{95.38} \pm 0.04$ | 95% |
| | | $92.06 \pm 0.38$ | $93.80 \pm 0.24$ | $93.92 \pm 0.16$ | $\mathbf{94.98} \pm 0.08$ | 97.5% |
| | | $89.49 \pm 0.20$ | $92.18 \pm 0.04$ | $92.50 \pm 0.07$ | $\mathbf{93.86} \pm 0.20$ | 99% |
| WideResNet-28-8 | $96.06 \pm 0.06$ | $95.49 \pm 0.21$ | $95.67 \pm 0.14$ | $95.81 \pm 0.10$ | $\mathbf{96.08} \pm 0.15$ | 90% |
| | | $94.92 \pm 0.13$ | $95.64 \pm 0.07$ | $95.55 \pm 0.12$ | $\mathbf{95.98} \pm 0.10$ | 95% |
| | | $94.11 \pm 0.19$ | $95.31 \pm 0.20$ | $95.11 \pm 0.07$ | $\mathbf{95.84} \pm 0.04$ | 97.5% |
| | | $92.04 \pm 0.11$ | $94.38 \pm 0.12$ | $94.10 \pm 0.12$ | $\mathbf{95.63} \pm 0.16$ | 99% |
| | | $74.50 \pm 2.23$ | $\star$ | $88.65 \pm 0.36$ | $\mathbf{91.76} \pm 0.18$ | 99.9% |

DPF on a large model (e.g. WideResNet-28-8) than on a smaller one (e.g. WideResNet-28-2). That is, DPF performs structural exploration more efficiently in larger parametric spaces.

Table 2: Top-1 test accuracy of SOTA DNNs on **CIFAR-10** for unstructured weight pruning via some simple pruning techniques. This table complements Table 1 and evaluates the performance of model compression under One-shot P+FT and Incremental, as well as how extra fine-tuning (FT) impact the performance of Incremental and our DPF. Note that One-shot P+FT prunes the dense model and uses extra fine-tuning itself. The Dense, Incremental and DPF train with the same number of epochs from scratch. The extra fine-tuning procedure considers the model checkpoint at the end of the normal training, uses the same number of training epochs (60 epochs in our case) with tuned optimizer and learning rate. Detailed hyperparameters tuning procedure refers to Appendix A.2.

| Model | Baseline on dense model | Methods | | | | | Target pr. ratio |
|---|---|---|---|---|---|---|---|
| | | One-shot + FT (H$^+$, 2015) | Incremental (ZG, 2017) | Incremental + FT | DPF | DPF + FT | |
| ResNet-20 | $92.48 \pm 0.20$ | $90.18 \pm 0.12$ | $90.55 \pm 0.38$ | $90.54 \pm 0.25$ | $90.88 \pm 0.07$ | $\mathbf{91.76} \pm 0.12$ | 90% |
| | | $86.91 \pm 0.16$ | $89.21 \pm 0.10$ | $89.24 \pm 0.28$ | $88.01 \pm 0.30$ | $\mathbf{90.34} \pm 0.31$ | 95% |
| ResNet-32 | $93.83 \pm 0.12$ | $91.72 \pm 0.15$ | $91.69 \pm 0.12$ | $91.76 \pm 0.14$ | $92.42 \pm 0.18$ | $\mathbf{92.61} \pm 0.11$ | 90% |
| | | $89.31 \pm 0.18$ | $90.86 \pm 0.17$ | $90.93 \pm 0.18$ | $90.94 \pm 0.35$ | $\mathbf{92.18} \pm 0.14$ | 95% |
| ResNet-56 | $94.51 \pm 0.20$ | $93.26 \pm 0.06$ | $93.14 \pm 0.23$ | $93.09 \pm 0.16$ | $\mathbf{93.95} \pm 0.11$ | $93.95 \pm 0.17$ | 90% |
| | | $91.61 \pm 0.07$ | $92.14 \pm 0.10$ | $92.50 \pm 0.25$ | $92.74 \pm 0.08$ | $\mathbf{93.25} \pm 0.15$ | 95% |

**ImageNet.** We compared DPF to other dynamic reparameterization methods as well as the strong Incremental baseline in Table 3. For both sparsity levels (80% and 90%), DPF shows a significant improvement of top-1 test accuracy with fewer or equal number of parameters.

Table 3: Top-1 test accuracy of **ResNet-50** on **ImageNet** for unstructured weight pruning. The # of parameters for the full model is 25.56 M. We used the results of DSR from Mostafa & Wang (2019) as we use the same (standard) training/data augmentation scheme for the same neural architecture. Note that different methods prune different types of layers and result in the different # of parameters for the same target pruning ratio. We also directly (and fairly) compare with the results of Incremental (Zhu & Gupta, 2017) reproduced and fine-tuned by Gale et al. (2019), where they consider layer-wise sparsity ratio and fine-tune both the sparsity warmup schedule and label-smoothing for better performance.

| | Top-1 accuracy | | | Top-5 accuracy | | | Pruning ratio | | |
|---|---|---|---|---|---|---|---|---|---|
| Method | Dense | Pruned | Difference | Dense | Pruned | Difference | Target | Reached | remaining # of params |
| Incremental (ZG, 2017) | 75.95 | 74.25 | -1.70 | 92.91 | 91.84 | -1.07 | 80% | 73.5% | **6.79 M** |
| DSR (MW, 2019) | 74.90 | 73.30 | -1.60 | 92.40 | 92.40 | **0** | 80% | 71.4% | 7.30 M |
| SM (DZ, 2019) | 75.95 | 74.59 | -1.36 | 92.91 | 92.37 | -0.54 | 80% | 72.4% | 7.06 M |
| DPF | 75.95 | 75.48 | **-0.47** | 92.91 | 92.59 | -0.32 | 80% | 73.5% | **6.79 M** |
| Incremental (Gale et al., 2019) | 76.69 | 75.58 | -1.11 | - | - | - | 80% | 79.9% | 5.15 M |
| DPF | 75.95 | 75.13 | **-0.82** | 92.91 | 92.52 | -0.39 | 80% | 79.9% | 5.15 M |
| Incremental (ZG, 2017) | 75.95 | 73.36 | -2.59 | 92.91 | 91.27 | -1.64 | 90% | 82.6% | **4.45 M** |
| DSR (MW, 2019) | 74.90 | 71.60 | -3.30 | 92.40 | 90.50 | -1.90 | 90% | 80.0% | 5.10 M |
| SM (DZ, 2019) | 75.95 | 72.65 | -3.30 | 92.91 | 91.26 | -1.65 | 90% | 82.0% | 4.59 M |
| DPF | 75.95 | 74.55 | **-1.44** | 92.91 | 92.13 | **-0.78** | 90% | 82.6% | **4.45 M** |

# 6 DISCUSSION

Besides the theoretical guarantees, a straightforward benefit of DPF over one-shot pruning in practice is its *fine-tuning free* training process. Figure 12 in the appendix (Section A.3.3) demonstrates the trivial computational overhead (considering the dynamic reparameterization cost) of involving DPF to train the model from scratch. Small number of hyperparameters compared to other dynamic reparameterization methods (e.g. DSR and SM) is another advantage of DPF and Figure 11 further studies how different setups of DPF impact the final performance. Notice also that for DPF, inference is done only at sparse models, an aspect that could be leveraged for more efficient computations.

**Empirical difference between one-shot pruning and DPF.** From the Figure 2 one can see that DPF tends to oscillate among several local minima, whereas one-shot pruning, even with fine tuning, converges to a single solution, which is not necessarily close to the optimum. We believe that the wider exploration of DPF helps to find a better local minima (which can be even further improved by fine-tuning, as shown in Table 2). We empirically analyzed how drastically the mask changes between each reparameterization step, and how likely it is for some pruned weight to become non-zero in the later stages of training. Figure 4 shows at what stage of the training each element of the final mask becomes fixed. For each epoch, we report how many mask elements were flipped starting from this epoch. As an example, we see that for sparsity ratio 95%, after epoch 157 (i.e. for 43 epochs left), only 5% of the mask elements were changing. This suggests that, except for a small percentage of weights that keep oscillating, the mask has converged early in the training. In the final epochs, the algorithm keeps improving accuracy, but the masks are only being fine-tuned. A similar mask convergence behavior can be found in Appendix (Figure 7) for training ResNet-20 on CIFAR-10.

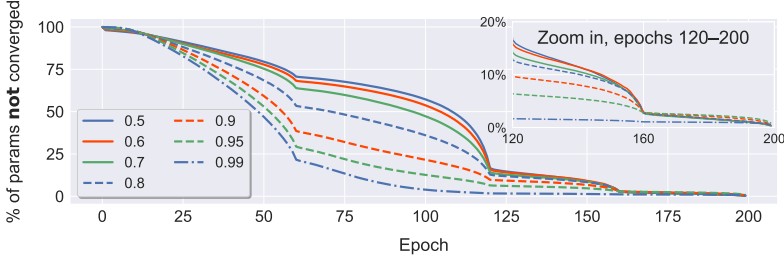

Figure 4: Convergence of the pruning mask $\mathbf{m}_t$ of DPF for different target sparsity levels (see legend). The $y$-axis represent the percentage of mask elements that still change **after** a certain epoch ($x$-axis). The illustrated example are from WideResNet-28-2 on CIFAR-10. We decayed the learning rate at 60,120,160 epochs.

**DPF does not find a lottery ticket.** The LT hypothesis (Frankle & Carbin, 2019) conjectures that for every desired sparsity level there exists a sparse submodel that can be trained to the same or better performance as the dense model. In Figure 5 we show that the mask found by DPF is not a LT, i.e., training the obtained sparse model from scratch does not recover the same performance. The (expensive) procedure proposed in Frankle & Carbin (2019); Frankle et al. (2019) finds different masks and achieves the same performance as DPF for mild sparsity levels, but DPF is much better for extremely sparse models (99% sparsity).

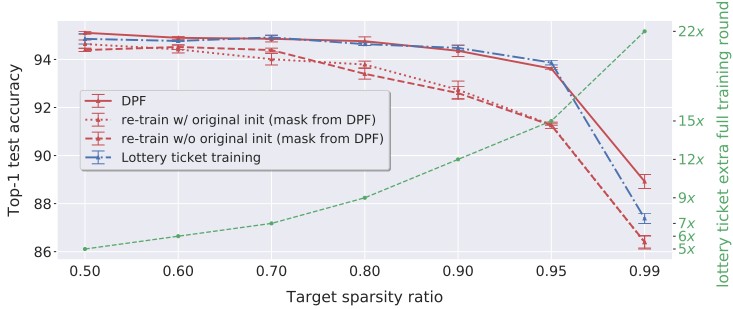

Figure 5: Top-1 test accuracy for different target sparsity levels (on WideResNet-28-2 with CIFAR-10, unstructured pruning). DPF reaches comparable accuracy than the LT training method (and better for 99% target sparsity), but involves much less computation (**right $y$-axis, green**). Training the sparse models found by DPF from scratch does not reach the same performance (hence our sparse models are not lottery tickets).

**Extension to structured pruning.** The current state-of-the-art dynamic reparameterization methods only consider unstructured weight pruning. Structured filter pruning[6] is either ignored (Bellec et al., 2018; Mocanu et al., 2018; Dettmers & Zettlemoyer, 2019) or shown to be challenging (Mostafa & Wang, 2019) even for the CIFAR dataset. In Figure 6 below we presented some preliminary results on CIFAR-10 to show that our DPF can also be applied to structured filter pruning schemes. DPF *outperforms the current filter-norm based state-of-the-art method for structured pruning (e.g. SFP (He et al., 2018)) by a noticeable margin.* Figure 16 in Appendix A.4.3 displays the transition procedure of the sparsity pattern (of different layers) for WideResNet-28-2 with different sparsity levels. DPF can be seen as a particular neural architecture search method, as it gradually learns to prune entire layers under the guidance of the feedback signal.

We followed the common experimental setup as mentioned in Section 5 with $\ell_2$ norm based filter selection criteria for structured pruning extension on CIFAR-10. We do believe a better filter selection scheme (Ye et al., 2018; He et al., 2019; Lym et al., 2019) could further improve the results but we leave this exploration for the future work.

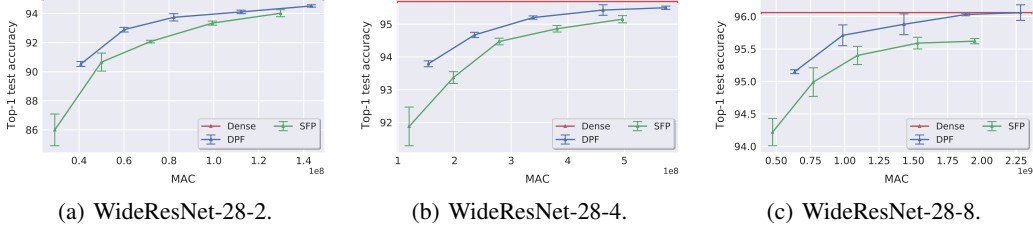

| (a) WideResNet-28-2. | (b) WideResNet-28-4. | (c) WideResNet-28-8. |

Figure 6: MAC v.s. top-1 test accuracy, for training WideResNet-28 (with different width) on CIFAR-10. The reported results are averaged over three runs. The WideResNet-28-2 has 216M MACs, WideResNet-28-4 has 848M MACs and WideResNet-28-8 has 3366M MACs. Other detailed information refers to the Appendix A.4.1, e.g., the # of params v.s. top-1 test accuracy in Figure 14, and the numerical test accuracy score in Table 6.

---

[6]Lym et al. (2019) consider structured filter pruning and reconfigure the large (but sparse) model to small (but dense) model during the training for better training efficiency. Note that they perform model update on a gradually reduced model space, and it is completely different from the dynamic reparameterization methods (e.g. DSR, SM and our scheme) that perform reparameterization under original (full) model space.

ACKNOWLEDGEMENTS

We acknowledge funding from SNSF grant 200021_175796, as well as a Google Focused Research Award.

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

## A  APPENDIX

### A.1  ALGORITHM

---

**Algorithm 1** *The detailed training procedure of* DPF.

---

**input:** uncompressed model weights $\mathbf{w} \in \mathbb{R}^d$, pruned weights: $\widetilde{\mathbf{w}}$, mask: $\mathbf{m} \in \{0,1\}^d$; reparametrization period: $p$; training iterations: $T$.
1: **for** $t = 0, \ldots, T$ **do**
2:     **if** $p \mid t$ **then**                                          ▷ trigger mask update, per default every $p = 16$ iterations
3:         compute mask $\mathbf{m} \leftarrow \mathbf{m}_t(\mathbf{w}_t)$  ▷ by arbitrary pruning scheme (e.g. unstructured magnitude pruning)
4:     **end if**
5:     $\widetilde{\mathbf{w}}_t \leftarrow \mathbf{m} \odot \mathbf{w}_t$                                          ▷ apply (precomputed) mask
6:     compute (mini-batch) gradient $\mathbf{g}(\widetilde{\mathbf{w}})$            ▷ forward/backward pass with pruned weights $\widetilde{\mathbf{w}}_t$
7:     $\mathbf{w}_{t+1} \leftarrow$ gradient update $\mathbf{g}(\widetilde{\mathbf{w}})$ to $\mathbf{w}_t$        ▷ via arbitrary optimizer (e.g. SGD with momentum)
8: **end for**
**output:** $\mathbf{w}_T$ and $\widetilde{\mathbf{w}}_T$

---

We trigger the mask update every $p = 16$ iterations (see also Figure 11) and we keep this parameter fixed throughout all experiments, independent of architecture or task.

We perform pruning across all neural network layers (no layer-wise pruning) using magnitude-based unstructured weight pruning (inherited from Han et al. (2015)). Pruning is applied to all convolutional layers while keeping the last fully-connected layer, biases and batch normalization layers dense.

We gradually increases the sparsity $s_t$ of the mask from 0 to the desired sparsity using the same scheduling as in Zhu & Gupta (2017); see Appendix A.2 below.

### A.2  IMPLEMENTATION DETAILS

We implemented our DPF in PyTorch (Paszke et al., 2017). All experiments were run on NVIDIA Tesla V100 GPUs. Sparse tensors in our implementation are respresented as the dense tenors multiplied by the corresponding binary masks.

**Datasets**   We evaluate all methods on the following standard image classification tasks:

- Image classification for CIFAR-10 (Krizhevsky & Hinton, 2009). Dataset consists of a training set of 50K and a test set of 10K color images of $32 \times 32$ pixels, as well as 10 target classes. We adopt the standard data augmentation and preprocessing scheme (He et al., 2016a; Huang et al., 2016).

- Image classification for ImageNet (Russakovsky et al., 2015). The ILSVRC 2012 classification dataset consists of 1.28 million images for training, and 50K for validation, with 1K target classes. We use ImageNet-1k (Deng et al., 2009) and adopt the same data preprocessing and augmentation scheme as in He et al. (2016a;b); Simonyan & Zisserman (2015).

**Gradual Pruning Scheme**   For Incremental baseline, we tuned their automated gradual pruning scheme $s_t = s_f + (s_i - s_f) \left(1 - \frac{t - t_0}{n \Delta t}\right)^3$ to gradually adjust the pruning sparsity ratio $s_t$ for $t \in \{t_0, \ldots, t_0 + n\Delta t\}$. That is, in our setup, we increased from an initial sparsity ratio $s_i = 0$ to the desired target model sparsity ratio $s_f$ over the epoch ($n$) when performing the second learning rate decay, from the training epoch $t_0 = 0$ and with pruning frequency $\Delta t = 1$ epoch. In our experiments, we used this gradual pruning scheme over different methods, except One-shot P+FT, SNIP, and the methods (DSR, SM) that have their own fine-tuned gradual pruning scheme.

**Hyper-parameters tuning procedure**   We grid-searched the optimal learning rate, starting from the range of $\{0.05, 0.10, 0.15, 0.20\}$. More precisely, we will evaluate a linear-spaced grid of learning rates. If the best performance was ever at one of the extremes of the grid, we would try new grid points so that the best performance was contained in the middle of the parameters.

We trained most of the methods by using mini-batch SGD with Nesterov momentum. For baselines involving fine-tuning procedure (e.g. Table 2), we grid-searched the optimal results by tuning the optimizers (i.e. mini-batch SGD with Nesterov momentum, or Adam) and the learning rates.

**The optimal hyper-parameters for DPF** The mini-batch size is fixed to 128 for CIFAR-10 and 1024 for ImageNet regardless of datasets and models.

For CIFAR-10, we trained ResNet-$a$ and VGG for 300 epochs and decayed the learning rate by 10 when accessing $50\%$ and $75\%$ of the total training samples (He et al., 2016a; Huang et al., 2017); and we trained WideResNet-$a$-$b$ as Zagoruyko & Komodakis (2016) for 200 epochs and decayed the learning rate by 5 when accessing $30\%$, $60\%$ and $80\%$ of the total training samples. The optimal learning rate for ResNet-$a$, WideResNet-$a$-$b$ and VGG are 0.2, 0.1 and 0.2 respectively; the corresponding weight decays are $1e-4$, $5e-4$ and $1e-4$ respectively.

For ImageNet training, we used the training scheme in Goyal et al. (2017) for 90 epochs, where we gradually warmup the learning rate from 0.1 to 0.4 and decayed learning rate by 10 at $30; 60; 80$ epochs. The used weight decay is $1e-4$.

## A.3 Additional Results for Unstructured Pruning

### A.3.1 Complete Results of Unstructured Pruning on CIFAR-10

Table 4 details the numerical results for training SOTA DNNs on CIFAR-10. Some results of it reconstruct the Table 1 and Figure 3.

Table 4: Top-1 test accuracy for training (compressed) SOTA DNNs on **CIFAR-10** from scratch. We considered unstructured pruning and the $\star$ indicates the method cannot converge. The results are averaged for three runs.

| Model | Baseline on dense model | Methods | | | | Target Pr. ratio |
| | | SNIP (L$^+$, 2019) | SM (DZ, 2019) | DSR (MW, 2019) | DPF | |
|---|---|---|---|---|---|---|
| VGG16-D | $93.74 \pm 0.13$ | $93.04 \pm 0.26$ | $93.59 \pm 0.17$ | - | $\mathbf{93.87 \pm 0.15}$ | 95% |
| ResNet-20 | $92.48 \pm 0.20$ | $91.10 \pm 0.22$ | $91.98 \pm 0.01$ | $92.00 \pm 0.19$ | $\mathbf{92.42 \pm 0.14}$ | 70% |
| ResNet-20 | $92.48 \pm 0.20$ | $90.53 \pm 0.27$ | $91.54 \pm 0.16$ | $91.78 \pm 0.28$ | $\mathbf{92.17 \pm 0.21}$ | 80% |
| ResNet-20 | $92.48 \pm 0.20$ | $88.50 \pm 0.13$ | $89.76 \pm 0.40$ | $87.88 \pm 0.04$ | $\mathbf{90.88 \pm 0.07}$ | 90% |
| ResNet-20 | $92.48 \pm 0.20$ | $84.91 \pm 0.25$ | $83.03 \pm 0.74$ | $\star$ | $\mathbf{88.01 \pm 0.30}$ | 95% |
| ResNet-32 | $93.83 \pm 0.12$ | $90.40 \pm 0.26$ | $91.54 \pm 0.18$ | $91.41 \pm 0.23$ | $\mathbf{92.42 \pm 0.18}$ | 90% |
| ResNet-32 | $93.83 \pm 0.12$ | $87.23 \pm 0.29$ | $88.68 \pm 0.22$ | $84.12 \pm 0.32$ | $\mathbf{90.94 \pm 0.35}$ | 95% |
| ResNet-56 | $94.51 \pm 0.20$ | $91.43 \pm 0.34$ | $92.73 \pm 0.21$ | $93.78 \pm 0.20$ | $\mathbf{93.95 \pm 0.11}$ | 90% |
| ResNet-56 | $94.51 \pm 0.20$ | $\star$ | $90.96 \pm 0.40$ | $92.57 \pm 0.09$ | $\mathbf{92.74 \pm 0.08}$ | 95% |
| WideResNet-28-2 | $95.01 \pm 0.04$ | $94.67 \pm 0.23$ | $94.73 \pm 0.16$ | $94.80 \pm 0.14$ | $\mathbf{95.11 \pm 0.06}$ | 50% |
| WideResNet-28-2 | $95.01 \pm 0.04$ | $94.47 \pm 0.19$ | $94.65 \pm 0.16$ | $94.98 \pm 0.07$ | $94.90 \pm 0.06$ | 60% |
| WideResNet-28-2 | $95.01 \pm 0.04$ | $94.29 \pm 0.22$ | $94.46 \pm 0.11$ | $94.80 \pm 0.15$ | $\mathbf{94.86 \pm 0.13}$ | 70% |
| WideResNet-28-2 | $95.01 \pm 0.04$ | $93.56 \pm 0.14$ | $94.17 \pm 0.12$ | $94.57 \pm 0.13$ | $\mathbf{94.76 \pm 0.18}$ | 80% |
| WideResNet-28-2 | $95.01 \pm 0.04$ | $92.58 \pm 0.22$ | $93.41 \pm 0.22$ | $93.88 \pm 0.08$ | $\mathbf{94.36 \pm 0.24}$ | 90% |
| WideResNet-28-2 | $95.01 \pm 0.04$ | $90.80 \pm 0.04$ | $92.24 \pm 0.14$ | $92.74 \pm 0.17$ | $\mathbf{93.62 \pm 0.05}$ | 95% |
| WideResNet-28-2 | $95.01 \pm 0.04$ | $83.45 \pm 0.38$ | $85.36 \pm 0.80$ | $\star$ | $\mathbf{88.92 \pm 0.29}$ | 99% |
| WideResNet-28-4 | $95.69 \pm 0.10$ | $95.42 \pm 0.05$ | $95.57 \pm 0.08$ | $95.67 \pm 0.07$ | $95.58 \pm 0.21$ | 70% |
| WideResNet-28-4 | $95.69 \pm 0.10$ | $95.24 \pm 0.07$ | $95.27 \pm 0.02$ | $95.49 \pm 0.04$ | $\mathbf{95.60 \pm 0.08}$ | 80% |
| WideResNet-28-4 | $95.69 \pm 0.10$ | $94.56 \pm 0.11$ | $95.01 \pm 0.05$ | $95.30 \pm 0.12$ | $\mathbf{95.65 \pm 0.14}$ | 90% |
| WideResNet-28-4 | $95.69 \pm 0.10$ | $93.62 \pm 0.17$ | $94.45 \pm 0.14$ | $94.63 \pm 0.08$ | $\mathbf{95.38 \pm 0.04}$ | 95% |
| WideResNet-28-4 | $95.69 \pm 0.10$ | $92.06 \pm 0.38$ | $93.80 \pm 0.24$ | $93.92 \pm 0.16$ | $\mathbf{94.98 \pm 0.08}$ | 97.5% |
| WideResNet-28-4 | $95.69 \pm 0.10$ | $89.49 \pm 0.20$ | $92.18 \pm 0.04$ | $92.50 \pm 0.07$ | $\mathbf{93.86 \pm 0.20}$ | 99% |
| WideResNet-28-8 | $96.06 \pm 0.06$ | $95.81 \pm 0.05$ | $95.92 \pm 0.12$ | $96.06 \pm 0.09$ | - | 70% |
| WideResNet-28-8 | $96.06 \pm 0.06$ | $95.86 \pm 0.10$ | $95.97 \pm 0.05$ | $96.05 \pm 0.12$ | - | 80% |
| WideResNet-28-8 | $96.06 \pm 0.06$ | $95.49 \pm 0.21$ | $95.67 \pm 0.14$ | $95.81 \pm 0.10$ | $\mathbf{96.08 \pm 0.15}$ | 90% |
| WideResNet-28-8 | $96.06 \pm 0.06$ | $94.92 \pm 0.13$ | $95.64 \pm 0.07$ | $95.55 \pm 0.12$ | $\mathbf{95.98 \pm 0.10}$ | 95% |
| WideResNet-28-8 | $96.06 \pm 0.06$ | $94.11 \pm 0.19$ | $95.31 \pm 0.20$ | $95.11 \pm 0.07$ | $\mathbf{95.84 \pm 0.04}$ | 97.5% |
| WideResNet-28-8 | $96.06 \pm 0.06$ | $92.04 \pm 0.11$ | $94.38 \pm 0.12$ | $94.10 \pm 0.12$ | $\mathbf{95.63 \pm 0.16}$ | 99% |
| WideResNet-28-8 | $96.06 \pm 0.06$ | $74.50 \pm 2.23$ | $\star$ | $88.65 \pm 0.36$ | $\mathbf{91.76 \pm 0.18}$ | 99.9% |

### A.3.2 Understanding the Training Dynamics and Lottery Ticket Effect

Figure 7 and Figure 8 complements Figure 4, and details the training dynamics (e.g. the converge of $\delta$ and masks) of DPF from the other aspect. Figure 9 compares the training dynamics between DPF

and Incremental (Zhu & Gupta, 2017), demonstrating the fact that our scheme enables a drastical reparameterization over the dense parameter space for better generalization performance.

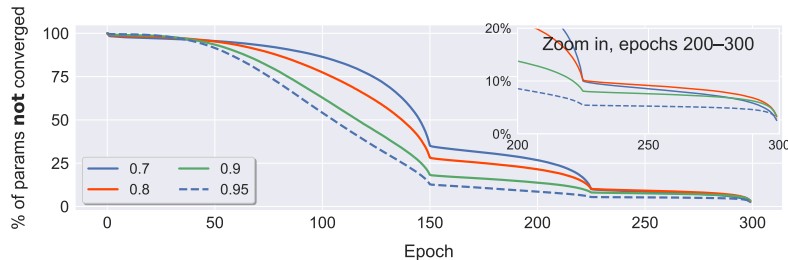

Figure 7: Convergence of the pruning mask $\mathbf{m}_t$ of DPF for different target sparsity levels (see legend). The $y$-axis represent the percentage of mask elements that still change **after** a certain epoch ($x$-axis). The illustrated example are from ResNet-20 on CIFAR-10. We decayed the learning rate at 150 and 225 epochs.

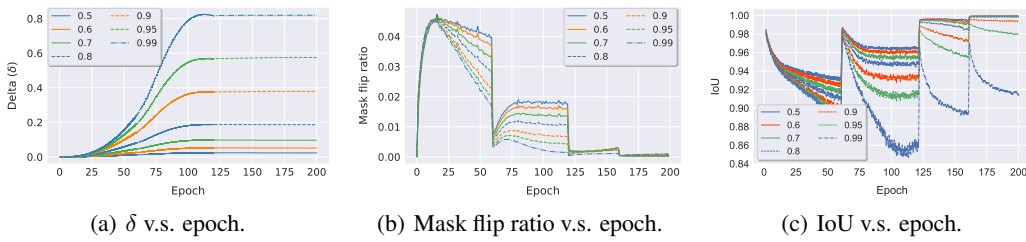

(a) $\delta$ v.s. epoch.      (b) Mask flip ratio v.s. epoch.      (c) IoU v.s. epoch.

Figure 8: Training dynamics of DPF (WideResNet-28-2 on CIFAR-10) for unstructured pruning with different sparsity ratios. IoU stands for Intersection over Union for the non-masked elements of two consecutive masks; the smaller value the more fraction of the masks will flip.

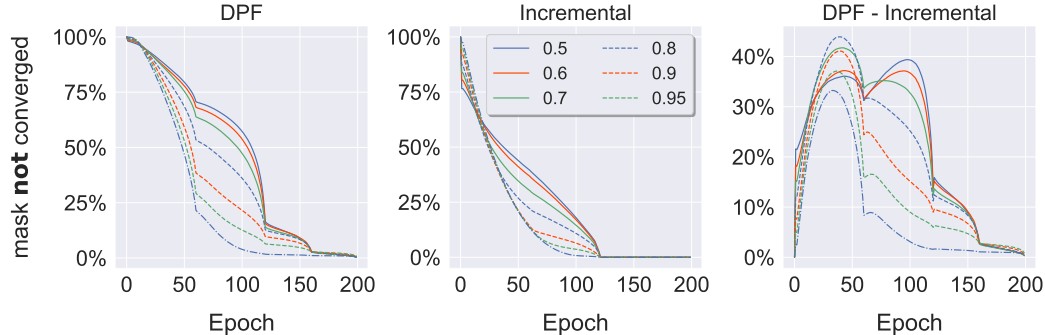

Figure 9: Convergence of the pruning mask $\mathbf{m}_t$ of DPF V.S. Incremental (Zhu & Gupta, 2017) for different target sparsity levels (see legend). The $y$-axis represent the percentage of mask elements that still change **after** a certain epoch ($x$-axis). The illustrated example are from WideResNet-28-2 on CIFAR-10. We decayed the learning rate at $60; 120; 160$ epochs. These two schemes use the same gradual warmup schedule (and hyper-parameters) for the pruning ratio during the training.

Figure 10 in addition to the Figure 5 (in the main text) further studies the lottery ticket hypothesis under different training budgets (same epochs or same total flops). The results of DPF also demonstrate the importance of training-time structural exploration as well as the corresponding implicit regularization effects. Note that we do not want to question the importance of the weight initialization or the existence of the lottery ticket. Instead, our DPF can provide an alternative training scheme to compress the model to an extremely high compression ratio without sacrificing the test accuracy, where most of the existing methods still meet severe quality loss (including Frankle & Carbin (2019); Liu et al. (2019); Frankle et al. (2019)).

### A.3.3 COMPUTATIONAL OVERHEAD AND THE IMPACT OF HYPER-PARAMETERS

In Figure 11, we evaluated the top-1 test accuracy of a compressed model trained by DPF under different setups, e.g., different reparameterization period $p$, different sparsity ratios, different mini-batch

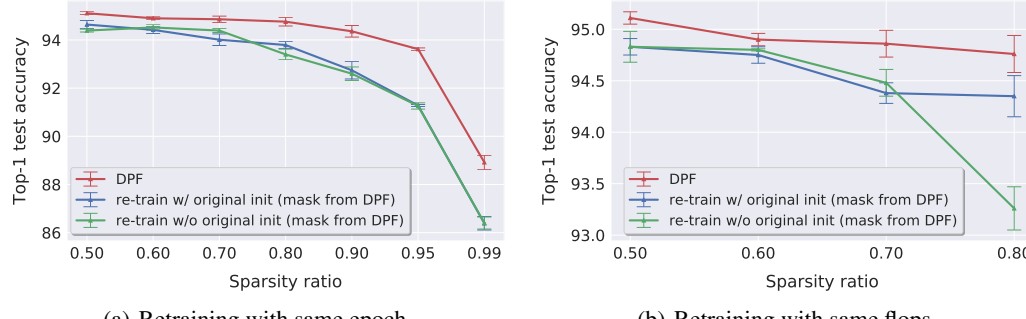

(a) Retraining with same epoch.  (b) Retraining with same flops.

Figure 10: Investigate the effect of lottery ticket for model compression (unstructured weight pruning for WideResNet28-2 on CIFAR-10). It complements the observations in Figure 5 by retraining the model for the same amount of computation budget (i.e. flops).

sizes, as well as whether layer-wise pruning or not. We can witness that the optimal reparameterization (i.e $p = 16$) is quite consistent over different sparsity ratios and different mini-batch sizes, and we used it in all our experiments. The global-wise unstructured weight pruning (instead of layer-wise weight pruning) allows our DPF more flexible to perform dynamic parameter reallocation, and thus can provide better results especially for more aggressive pruning sparsity ratios. However, we also need to note that, for the same number of compressed parameters (layerwise or globalwise unstructured weight pruning), using global-wise pruning leads to a slight increase in the amount of MACs, as illustrated Table 5.

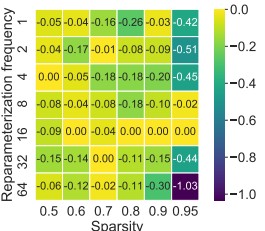 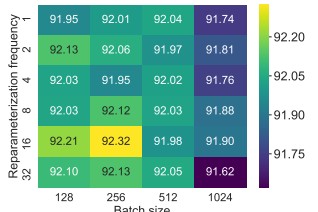 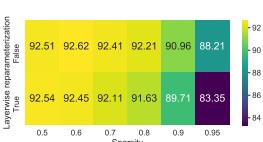

(a) Reparameterization period vs. sparsity ratio. The heatmap value is the test accuracy minus the best accuracy of the corresponding sparsity. We use mini-batch size 128 w/o layerwise reparameterization.

(b) Reparameterization period vs. mini-batch size. The heatmap displays the test accuracy. We use sparsify ratio 0.80.

(c) Reparameterization scheme (whether layerwise) vs. sparsity ratio. The heatmap displays the test accuracy. We use mini-batch size 128 w/ reparameterization period $p = 16$.

Figure 11: Investigate how the reparameterization period/scheme and mini-batch size impact the generalization performance (test top-1 accuracy), for dynamically training (and reparameterizing) a compressed model from scratch (ResNet-20 with CIFAR-10).

Table 5: Investigate how the reparameterization scheme (layer-wise or not) impact the MACs (for the same number of compressed parameters), for using DPF on ResNet-20 with CIFAR-10.

| Target sparsity | 50% | 60% | 70% | 80% | 90% | 95% |
|---|---|---|---|---|---|---|
| w/o layerwise reparameterization | 22.80M | 19.13M | 15.18M | 11.12M | 6.54M | 4.02M |
| w/ layerwise reparameterization | 20.99M | 16.91M | 12.83M | 8.75M | 4.67M | 2.63M |

Figure 12 demonstrates the trivial computational overhead of involving DPF to gradually train a compressed model (ResNet-50) from scratch (on ImageNet). Note that we evaluated the introduced reparameterization cost for dynamic pruning, which is independent of (potential) significant system speedup brought by the extreme high model sparsity. Even though our work did not estimate the practical speedup, we do believe we can have a similar training efficiency as the values reported in Dettmers & Zettlemoyer (2019).

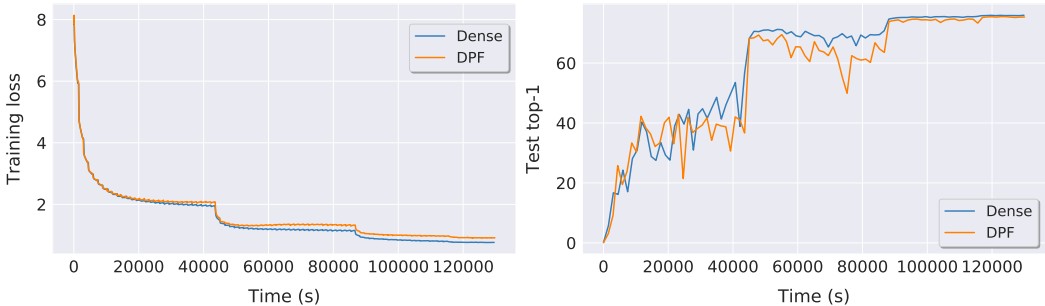

Figure 12: The learning curve of our DPF (with unstructured magnitude pruning) and the standard mini-batch SGD for training ResNet50 on ImageNet. Our proposed DPF has trivial computational overhead. We trained ResNet50 on 4 NVIDIA V100 GPUs with 1024 mini-batch size. The target sparsity ratio is 80%.

### A.3.4 IMPLICIT NEURAL ARCHITECTURE SEARCH

DPF can provide effective training-time structural exploration or even implicit neural network search. Figure 13 below demonstrates that for the same pruned model size (i.e. any point in the $x$-axis), we can always perform "architecture search" to get a better (in terms of generalization) pruned model, from a larger network (e.g. WideResNet-28-8) rather than the one searched from a relatively small network (e.g. WideResNet-28-4).

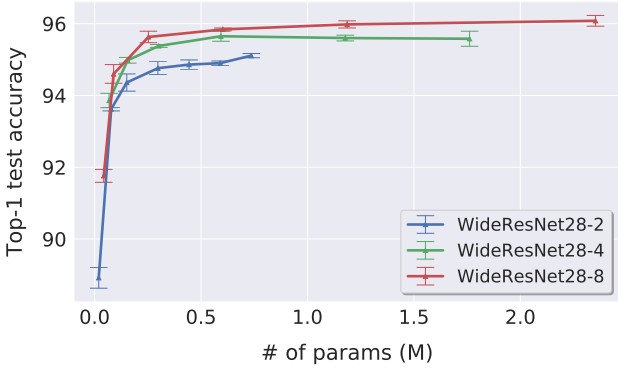

Figure 13: Test top-1 accuracy vs. the compressed model size, for training WideResNet-28 (with different widths) on CIFAR-10. The compressed model is searched from WideResNet-28 (fixed depth) with different width (number of filters per layer).

### A.4 ADDITIONAL RESULTS FOR STRUCTURED PRUNING

### A.4.1 GENERALIZATION PERFORMANCE FOR CIFAR-10

Figure 14 complements the results of structured pruning in the main text (Figure 6), and Table 6 details the numerical results presented in both of Figure 6 and Figure 14.

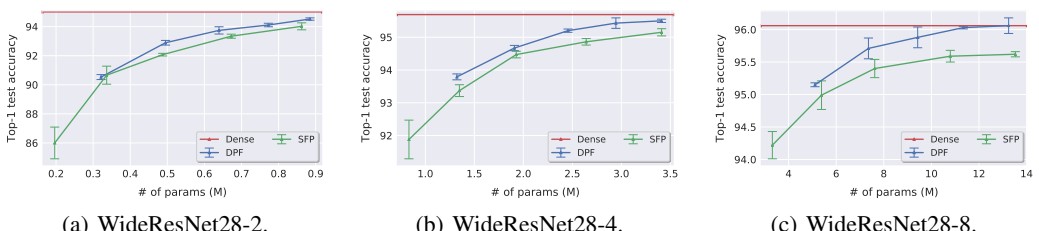

(a) WideResNet28-2.  (b) WideResNet28-4.  (c) WideResNet28-8.

Figure 14: # of params v.s. top-1 test accuracy, for training WideResNet28 (with different width) on CIFAR-10. Structured filter-wise pruning is used here. The reported results are averaged over three runs.

Table 6: Performance evaluation of DPF (and other baseline methods) for training (Wide)ResNet variants on CIFAR-10. We use the norm-based criteria for filter selection (as in SFP (He et al., 2018)) to estimate the output channel-wise pruning threshold. We follow the gradual pruning warmup scheme (as in Zhu & Gupta (2017)) from 0 epoch to the epoch when performing the second learning rate decay. Note that SFP prunes filters within the layer by a given ratio while our DPF prunes filters across layers. Due to the difference between filters for different layers, the # of parameters pruned by DPF might slight different from the one pruned by SFP. The pruning ratio refers to either prune filters within the layer or across the layers.

| | | Methods | | |
| Model | Baseline on dense model | SFP (H$^+$, 2018) | DPF | Target Pr. ratio |
| --- | --- | --- | --- | --- |
| ResNet-20 | $92.48 \pm 0.20$ | $92.18 \pm 0.31$ | $\mathbf{92.54} \pm 0.07$ | 10% |
| ResNet-20 | $92.48 \pm 0.20$ | $91.12 \pm 0.20$ | $\mathbf{91.90} \pm 0.06$ | 20% |
| ResNet-20 | $92.48 \pm 0.20$ | $90.32 \pm 0.25$ | $\mathbf{91.07} \pm 0.40$ | 30% |
| ResNet-20 | $92.48 \pm 0.20$ | $89.60 \pm 0.46$ | $\mathbf{90.28} \pm 0.26$ | 40% |
| ResNet-32 | $93.52 \pm 0.13$ | $92.07 \pm 0.22$ | $\mathbf{92.18} \pm 0.16$ | 30% |
| ResNet-32 | $93.52 \pm 0.13$ | $91.14 \pm 0.45$ | $\mathbf{91.50} \pm 0.21$ | 40% |
| ResNet-56 | $94.51 \pm 0.20$ | $93.99 \pm 0.27$ | $\mathbf{94.53} \pm 0.13$ | 30% |
| ResNet-56 | $94.51 \pm 0.20$ | $93.57 \pm 0.16$ | $\mathbf{94.03} \pm 0.38$ | 40% |
| WideResNet-28-2 | $95.01 \pm 0.04$ | $94.02 \pm 0.24$ | $\mathbf{94.52} \pm 0.08$ | 40% |
| WideResNet-28-2 | $95.01 \pm 0.04$ | $93.34 \pm 0.14$ | $\mathbf{94.11} \pm 0.12$ | 50% |
| WideResNet-28-2 | $95.01 \pm 0.04$ | $92.07 \pm 0.09$ | $\mathbf{93.74} \pm 0.25$ | 60% |
| WideResNet-28-2 | $95.01 \pm 0.04$ | $90.66 \pm 0.62$ | $\mathbf{92.89} \pm 0.16$ | 70% |
| WideResNet-28-2 | $95.01 \pm 0.04$ | $86.00 \pm 1.09$ | $\mathbf{90.53} \pm 0.17$ | 80% |
| WideResNet-28-4 | $95.69 \pm 0.10$ | $95.15 \pm 0.11$ | $\mathbf{95.50} \pm 0.05$ | 40% |
| WideResNet-28-4 | $95.69 \pm 0.10$ | $94.86 \pm 0.10$ | $\mathbf{95.43} \pm 0.16$ | 50% |
| WideResNet-28-4 | $95.69 \pm 0.10$ | $94.47 \pm 0.10$ | $\mathbf{95.20} \pm 0.05$ | 60% |
| WideResNet-28-4 | $95.69 \pm 0.10$ | $93.37 \pm 0.18$ | $\mathbf{94.67} \pm 0.08$ | 70% |
| WideResNet-28-4 | $95.69 \pm 0.10$ | $91.88 \pm 0.59$ | $\mathbf{93.79} \pm 0.09$ | 80% |
| WideResNet-28-8 | $96.06 \pm 0.06$ | $95.62 \pm 0.04$ | $\mathbf{96.06} \pm 0.12$ | 40% |
| WideResNet-28-8 | $96.06 \pm 0.06$ | $95.59 \pm 0.09$ | $\mathbf{96.03} \pm 0.02$ | 50% |
| WideResNet-28-8 | $96.06 \pm 0.06$ | $95.40 \pm 0.14$ | $\mathbf{95.88} \pm 0.16$ | 60% |
| WideResNet-28-8 | $96.06 \pm 0.06$ | $94.99 \pm 0.22$ | $\mathbf{95.71} \pm 0.16$ | 70% |
| WideResNet-28-8 | $96.06 \pm 0.06$ | $94.22 \pm 0.21$ | $\mathbf{95.15} \pm 0.03$ | 80% |

### A.4.2 UNDERSTANDING THE LOTTERY TICKET EFFECT

Similar to the observations in Section 6 (for unstructured pruning), Figure 15 instead considers structured pruning and again we found DPF does not find a lottery ticket. The superior generalization performance of DPF cannot be explained by the found mask or the weight initialization scheme.

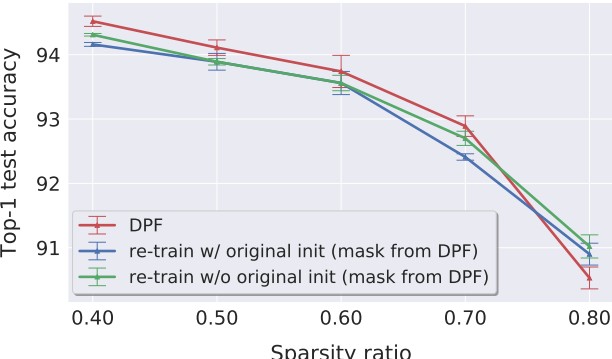

Figure 15: Investigate the effect of lottery ticket for model compression (WideResNet28-2 with CIFAR-10) for structured pruning. We retrained the model with the mask from the model trained by DPF, by using the same epoch budget.

### A.4.3 MODEL SPARSITY VISUALIZATION

Figure 16 below visualizes the model sparsity transition patterns for different model sparsity levels under the structured pruning. We can witness that due to the presence of residual connection, DPF gradually learns to prune the entire residual blocks.

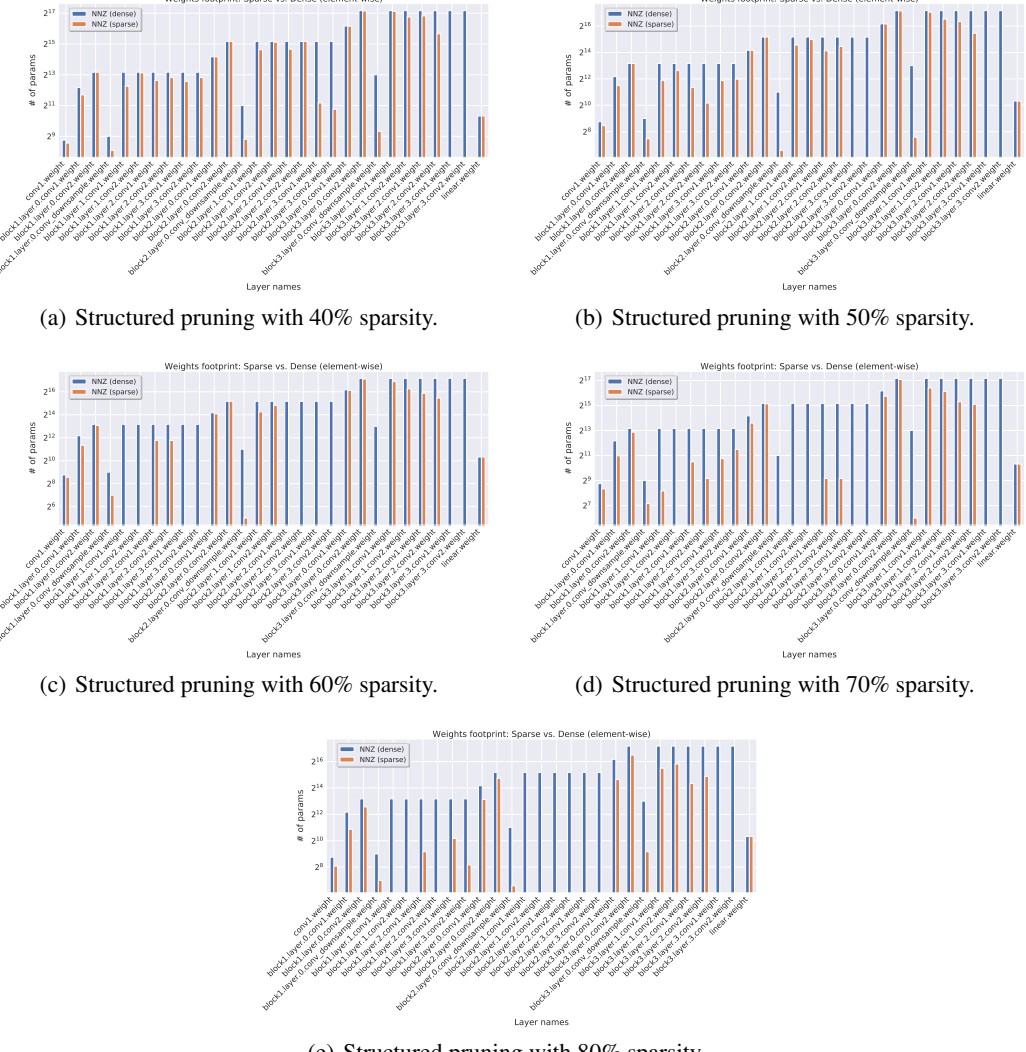

Figure 16: The element-wise sparsity of each layer for WideResNet28-2 (trained on CIFAR-10 via DPF), under different structured pruning sparsity ratios. DPF for model compression (with structured pruning) performs implicit as a neural architecture search.

# B MISSING PROOFS

In this section we present the proofs for the claims in Section 4.

First, we give the proof for the stongly convex case. Here we follow Lacoste-Julien et al. (2012) for the general structure, combined with estimates from the error-feedback framework (Stich et al., 2018; Stich & Karimireddy, 2019) to control the pruning errors.

*Proof of Theorem 4.1.* By definition of (DPF), $\mathbf{w}_{t+1} = \mathbf{w}_t - \gamma_t \mathbf{g}(\widetilde{\mathbf{w}}_t)$, hence,

$$
\begin{aligned}
\mathbb{E}\left[\|\mathbf{w}_{t+1} - \mathbf{w}^\star\|^2 \mid \mathbf{w}_t\right] &= \|\mathbf{w}_t - \mathbf{w}^\star\|^2 - 2\gamma_t \langle \mathbf{w}_t - \mathbf{w}^\star, \mathbb{E}\mathbf{g}(\widetilde{\mathbf{w}}_t)\rangle + \gamma_t^2 \mathbb{E}\|\mathbf{g}(\widetilde{\mathbf{w}}_t)\|^2 \\
&\leq \|\mathbf{w}_t - \mathbf{w}^\star\|^2 - 2\gamma_t \langle \mathbf{w}_t - \mathbf{w}^\star, \nabla f(\widetilde{\mathbf{w}}_t)\rangle + \gamma_t^2 G^2 \\
&= \|\mathbf{w}_t - \mathbf{w}^\star\|^2 - 2\gamma_t \langle \widetilde{\mathbf{w}}_t - \mathbf{w}^\star, \nabla f(\widetilde{\mathbf{w}}_t)\rangle + \gamma_t^2 G^2 \\
&\quad + 2\gamma_t \langle \widetilde{\mathbf{w}}_t - \mathbf{w}_t, \nabla f(\widetilde{\mathbf{w}}_t)\rangle .
\end{aligned}
$$

By strong convexity,

$$
-2\langle \widetilde{\mathbf{w}}_t - \mathbf{w}^\star, \nabla f(\widetilde{\mathbf{w}}_t)\rangle \leq -\mu\|\widetilde{\mathbf{w}}_t - \mathbf{w}^\star\|^2 - 2\left(f(\widetilde{\mathbf{w}}_t) - f(\mathbf{w}^\star)\right),
$$

and with $\|\mathbf{a} + \mathbf{b}\|^2 \leq 2\|\mathbf{a}\|^2 + 2\|\mathbf{b}\|^2$ further

$$
-\|\widetilde{\mathbf{w}}_t - \mathbf{w}^\star\|^2 \leq -\frac{1}{2}\|\mathbf{w}_t - \mathbf{w}^\star\|^2 + \|\mathbf{w}_t - \widetilde{\mathbf{w}}_t\|^2
$$

and with $\langle \mathbf{a}, \mathbf{b}\rangle \leq \frac{1}{2\alpha}\|\mathbf{a}\|^2 + \frac{\alpha}{2}\|\mathbf{b}\|^2$ for $\mathbf{a}, \mathbf{b} \in \mathbb{R}^d$ and $\alpha > 0$,

$$
\begin{aligned}
2\langle \widetilde{\mathbf{w}}_t - \mathbf{w}_t, \nabla f(\widetilde{\mathbf{w}}_t)\rangle &\leq 2L\|\widetilde{\mathbf{w}}_t - \mathbf{w}_t\|^2 + \frac{1}{2L}\|\nabla f(\widetilde{\mathbf{w}}_t)\|^2 \\
&= 2L\|\widetilde{\mathbf{w}}_t - \mathbf{w}_t\|^2 + \frac{1}{2L}\|\nabla f(\widetilde{\mathbf{w}}_t) - \nabla f(\mathbf{w}^\star)\|^2 \\
&\leq 2L\|\widetilde{\mathbf{w}}_t - \mathbf{w}_t\|^2 + f(\widetilde{\mathbf{w}}_t) - f(\mathbf{w}^\star),
\end{aligned}
$$

where the last inequality is a consequence of $L$-smoothness. Combining all these inequalities yields

$$
\begin{aligned}
\mathbb{E}\left[\|\mathbf{w}_{t+1} - \mathbf{w}^\star\|^2 \mid \mathbf{w}_t\right] &\leq \left(1 - \frac{\mu\gamma_t}{2}\right)\|\mathbf{w}_t - \mathbf{w}^\star\|^2 - \gamma_t\left(f(\widetilde{\mathbf{w}}_t) - f(\mathbf{w}^\star)\right) + \gamma_t^2 G^2 \\
&\quad + \gamma_t(2L + \mu)\|\widetilde{\mathbf{w}}_t - \mathbf{w}_t\|^2 \\
&\leq \left(1 - \frac{\mu\gamma_t}{2}\right)\|\mathbf{w}_t - \mathbf{w}^\star\|^2 - \gamma_t\left(f(\widetilde{\mathbf{w}}_t) - f(\mathbf{w}^\star)\right) + \gamma_t^2 G^2 \\
&\quad + 3\gamma_t L\|\widetilde{\mathbf{w}}_t - \mathbf{w}_t\|^2 .
\end{aligned}
$$

as $\mu \leq L$. Hence, by rearranging and multiplying with a weight $\lambda_t > 0$:

$$
\begin{aligned}
\lambda_t \mathbb{E}\left(f(\widetilde{\mathbf{w}}_t) - f(\mathbf{w}^\star)\right) &\leq \frac{\lambda_t(1 - \mu\gamma_t/2)}{\gamma_t}\mathbb{E}\|\mathbf{w}_t - \mathbf{w}^\star\|^2 - \frac{\lambda_t}{\gamma_t}\mathbb{E}\|\mathbf{w}_{t+1} - \mathbf{w}^\star\|^2 + \gamma_t\lambda_t G^2 \\
&\quad + 3\lambda_t L\mathbb{E}\|\widetilde{\mathbf{w}}_t - \mathbf{w}_t\|^2 .
\end{aligned}
$$

By plugging in the learning rate, $\gamma_t = \frac{4}{\mu(t+2)}$ and setting $\lambda_t = (t+1)$ we obtain

$$
\begin{aligned}
\lambda_t \mathbb{E}\left(f(\widetilde{\mathbf{w}}_t) - f(\mathbf{w}^\star)\right) &\leq \frac{\mu}{4}\left[t(t+1)\mathbb{E}\|\mathbf{w}_t - \mathbf{w}^\star\|^2 - (t+1)(t+2)\mathbb{E}\|\mathbf{w}_{t+1} - \mathbf{w}^\star\|^2\right] \\
&\quad + \frac{4(t+1)}{\mu(t+2)}G^2 + 3(t+1)L\mathbb{E}\|\widetilde{\mathbf{w}}_t - \mathbf{w}_t\|^2 .
\end{aligned}
$$

By summing from $t = 0$ to $t = T$ these $\lambda_t$-weighted inequalities, we obtain a telescoping sum:

$$
\begin{aligned}
\sum_{t=0}^{T} \lambda_t \mathbb{E}\left(f(\widetilde{\mathbf{w}}_t) - f(\mathbf{w}^\star)\right) &\leq \frac{\mu}{4}\left[0 - (T+1)(T+2)\mathbb{E}\|\mathbf{w}_{t+1} - \mathbf{w}^\star\|^2\right] + \frac{4(T+1)}{\mu}G^2 \\
&\quad + 3L\sum_{t=0}^{T}\lambda_t \mathbb{E}\|\widetilde{\mathbf{w}}_t - \mathbf{w}_t\|^2 \\
&\leq \frac{4(T+1)}{\mu}G^2 + 3L\sum_{t=0}^{T}\lambda_t \mathbb{E}\|\widetilde{\mathbf{w}}_t - \mathbf{w}_t\|^2 .
\end{aligned}
$$

Hence, for $\Lambda_T := \sum_{t=0}^{T} \lambda_t = \frac{(T+1)(T+2)}{2}$,

$$\frac{1}{\Lambda_T} \sum_{t=0}^{T} \lambda_t \mathbb{E} \left( f(\widetilde{\mathbf{w}}_t) - f(\mathbf{w}^\star) \right) \le \frac{4(T+1)}{\mu \Lambda_T} G^2 + \frac{3L}{\Lambda_T} \sum_{t=0}^{T} \lambda_t \mathbb{E} \left\| \widetilde{\mathbf{w}}_t - \mathbf{w}_t \right\|^2$$

$$= \mathcal{O} \left( \frac{G^2}{\mu T} + \frac{L}{\Lambda_T} \sum_{t=0}^{T} \lambda_t \mathbb{E} \left\| \widetilde{\mathbf{w}}_t - \mathbf{w}_t \right\|^2 \right) .$$

Finally, using $\| \widetilde{\mathbf{w}}_t - \mathbf{w}_t \|^2 = \delta_t \| \mathbf{w}_t \|^2$ by (1), shows the theorem. $\square$

Before giving the proof of Theorem 4.2, we first give a justification for the remark just below Theorem 4.1 on the one-shot pruning of the final iterate.

We have by $L$-smoothness and $\langle \mathbf{a}, \mathbf{b} \rangle \le \frac{1}{2\alpha} \| \mathbf{a} \|^2 + \frac{\alpha}{2} \| \mathbf{b} \|^2$ for $\mathbf{a}, \mathbf{b} \in \mathbb{R}^d$ and $\alpha > 0$ for any iterate $\mathbf{w}_t$:

$$f(\widetilde{\mathbf{w}}_t) - f(\mathbf{w}^\star) \le f(\mathbf{w}_t) - f(\mathbf{w}^\star) + \langle \nabla f(\mathbf{w}_t), \widetilde{\mathbf{w}}_t - \mathbf{w}_t \rangle + \frac{L}{2} \| \widetilde{\mathbf{w}}_t - \mathbf{w}_t \|^2$$

$$\le f(\mathbf{w}_t) - f(\mathbf{w}^\star) + \frac{1}{2L} \| \nabla f(\mathbf{w}_t) \|^2 + L \| \widetilde{\mathbf{w}}_t - \mathbf{w}_t \|^2$$

$$\le 2(f(\mathbf{w}_t) - f(\mathbf{w}^\star)) + \delta_t L \| \mathbf{w}_t \|^2 . \tag{4}$$

Furthermore, again by $L$-smoothness,

$$f(\mathbf{w}_T) - f(\mathbf{w}^\star) \le \frac{L}{2} \| \mathbf{w}_T - \mathbf{w}^\star \|^2 = \mathcal{O} \left( \frac{LG^2}{\mu^2 T} \right)$$

as standard SGD analysis gives the estimate $\mathbb{E} \| \mathbf{w}_T - \mathbf{w}^\star \|^2 = \mathcal{O} \left( \frac{G^2}{\mu^2 T} \right)$, see e.g. Lacoste-Julien et al. (2012). Combining these two estimates (with $\mathbf{w}_t = \mathbf{w}_T$) shows the claim.

Furthermore, we also claimed that also the dense model converges to a neighborhood of optimal solution. This follows by $L$-smoothness and (4): For any fixed model $\mathbf{w}_t$ we have the estimate (4), hence for a randomly chosen (dense) model $\mathbf{u}$ (from the same distribution as the sparse model in Theorem 4.1) we have

$$\mathbb{E} f(\mathbf{u}) - f(\mathbf{w}^\star) \overset{(4)}{\le} 2\mathbb{E} \left[ f(\widetilde{\mathbf{u}}) - f(\mathbf{w}^\star) \right] + L\mathbb{E} \left[ \delta_t \| \mathbf{w}_t \|^2 \right] \overset{\text{(Thm 4.1)}}{=} \mathcal{O} \left( \frac{G^2}{\mu T} + L\mathbb{E} \left[ \delta_t \| \mathbf{w}_t \|^2 \right] \right) .$$

Lastly, we give the proof of Theorem 4.2, following Karimireddy et al. (2019).

*Proof of Theorem 4.2.* By smoothness, and $\langle \mathbf{a}, \mathbf{b} \rangle \le \frac{1}{2} \| \mathbf{a} \|^2 + \frac{1}{2} \| \mathbf{b} \|^2$ for $\mathbf{a}, \mathbf{b} \in \mathbb{R}^d$,

$$\mathbb{E} \left[ f(\mathbf{w}_{t+1}) \mid \mathbf{w}_t \right] \le f(\mathbf{w}_t) - \gamma \langle \nabla f(\mathbf{w}_t), \mathbb{E} \mathbf{g}(\widetilde{\mathbf{w}}_t) \rangle + \gamma^2 \frac{L}{2} \mathbb{E} \| \mathbf{g}(\widetilde{\mathbf{w}}_t) \|^2$$

$$\le f(\mathbf{w}_t) - \gamma \langle \nabla f(\mathbf{w}_t), \nabla f(\widetilde{\mathbf{w}}_t) \rangle + \gamma^2 \frac{LG^2}{2}$$

$$= f(\mathbf{w}_t) - \gamma \langle \nabla f(\widetilde{\mathbf{w}}_t), \nabla f(\widetilde{\mathbf{w}}_t) \rangle + \gamma^2 \frac{LG^2}{2}$$

$$+ \gamma \langle \nabla f(\widetilde{\mathbf{w}}_t) - \nabla f(\mathbf{w}_t), \nabla f(\widetilde{\mathbf{w}}_t) \rangle$$

$$\le f(\mathbf{w}_t) - \gamma \| \nabla f(\widetilde{\mathbf{w}}_t) \|^2 + \gamma^2 \frac{LG^2}{2}$$

$$+ \frac{\gamma}{2} \| \nabla f(\widetilde{\mathbf{w}}_t) - \nabla f(\mathbf{w}_t) \|^2 + \frac{\gamma}{2} \| \nabla f(\widetilde{\mathbf{w}}_t) \|^2$$

$$\le f(\mathbf{w}_t) - \frac{\gamma}{2} \| \nabla f(\widetilde{\mathbf{w}}_t) \|^2 + \gamma^2 \frac{LG^2}{2} + \frac{\gamma L^2}{2} \| \mathbf{w}_t - \widetilde{\mathbf{w}}_t \|^2 ,$$

and by rearranging

$$\mathbb{E} \| \nabla f(\widetilde{\mathbf{w}}_t) \|^2 \le \frac{2}{\gamma} \left[ \mathbb{E} f(\mathbf{w}_t) - \mathbb{E} f(\mathbf{w}_{t+1}) \right] + \gamma LG^2 + L^2 \mathbb{E} \| \mathbf{w}_t - \widetilde{\mathbf{w}}_t \|^2 .$$

Summing these inequalities from $t = 0$ to $t = T$ gives

$$\frac{1}{T+1} \sum_{t=0}^{T} \mathbb{E} \left\| \nabla f(\widetilde{\mathbf{w}}_t) \right\|^2 \leq \frac{2}{\gamma(T+1)} \sum_{t=0}^{T} \left( \mathbb{E}\left[ f(\mathbf{w}_t) \right] - \mathbb{E}\left[ f(\mathbf{w}_{t+1}) \right] \right) + \gamma L G^2$$

$$+ \frac{L^2}{T+1} \sum_{t=0}^{T} \mathbb{E} \left\| \mathbf{w}_t - \widetilde{\mathbf{w}}_t \right\|^2$$

$$\leq \frac{2 \left( f(\mathbf{w}_0) - f(\mathbf{w}^\star) \right)}{\gamma(T+1)} + L\gamma G^2 + \frac{L^2}{T+1} \sum_{t=0}^{T} \mathbb{E} \left\| \mathbf{e}_t \right\|^2 .$$

Finally, using $\left\| \widetilde{\mathbf{w}}_t - \mathbf{w}_t \right\|^2 = \delta_t \left\| \mathbf{w}_t \right\|^2$ by (1), and plugging in the stepsize $\gamma$ that minimizes the right hand side shows the claim. $\qquad \square$

