# OpenReview forum: "Dynamic Model Pruning with Feedback"
_ICLR.cc/2020/Conference — Accept (Poster)_

### Official Review · AnonReviewer1 · 2019-10-18
**Official Blind Review #1**

**Rating:** 6

**Review:**

This work proposes a simple pruning method that dynamically sparsifies the network during training. This is achieved by performing at fixed intervals magnitude based pruning for either individual weights or entire neurons. While similar methods have been explored before, this work proposes a slight twist; instead of updating the weights of the model by following the gradient of the parameters of the dense model, they update the parameters of the dense model according to the gradients of the sparse model. Essentially, this corresponds to a variant of the straight-through estimator [1], where in the forward pass we evaluate the compressed model, but in the backward pass we update the model as if the compression didn’t take place. The authors argue that this process allows for ``feedback” in the pruning mechanism, as the pruned weights still receive gradient updates hence they can be ``re-activated” at later stages of training. They then provide a convergence analysis about the optimization procedure with such a gradient, and show that for strongly convex functions the method converges in the vicinity of the global optimum, whereas for non-convex functions it converges to the neighbourhood of a stationary point. Finally, the authors perform extensive experimental evaluation and show that their method is better than the baselines that they considered.

This work is in general well written and conveys the main idea in an effective manner.  It is also a timely contribution as sparse models / compression are important topics for the deep learning community. The overall method seems simple to implement, doesn’t introduce too many hyper-parameters and seem to work very well. For this reason I tend towards recommending for acceptance, provided that the authors address /comment on a couple of issues I found in the draft.

More specifically:
- The connection to the error feedback is kind of loose and not well explained. After skimming Karimireddy et al. I noticed that 1. it evaluates the gradient at a point (i.e. the current estimate of the parameters), 2. compresses said gradient, 3. updates the parameters while maintaining the difference of the original w.r.t. the compressed gradient. In this sense, it seems a bit different that DPF, as your notation at the first equation of page 4 implies that you take the gradient of a different point, i.e. w_t + e_t instead of w_t. I believe that expanding a bit more about the connection would help in making the manuscript more clear.
- There seems to be a typo / error on your definition of an m-strongly convex function at the “convergence of Convex functions” paragraph. I believe it should be <\nabla f(v), w-v> <= f(w) - f(v) - 0.5 m ||w - v||^2, instead of <\nabla f(w), w-v> <= f(w) - f(v) - 0.5 m ||w - v||^2.
- The proposed gradient estimator seems to be an instance of the STE [1] estimator, that, as the authors mention, has been using at the Binary Connect algorithm. It would be interesting to see some more discussion about this similarity perhaps also expanding upon recent work that discusses the STE gradient as a form of coarse gradient [2].
- At section 5.2 the authors mention that “dynamic pruning methods, and in particular DPF, work on a different paradigm, and can still heavily benefit from fine-tuning”. This claim seems to contradict the results at Figure 4; there it seems that the masks have “converged” in the later stages of training, hence one could argue that the fine-tuning already happens thus it wouldn’t benefit DPF. I believe it would be interesting if the authors provide a similar plot as the one in Figure 4 but rather for the ResNet-20 network on CIFAR 10 (which seems to benefit heavily from FT). Do the masks still settle at the end of training (as it was the case for WideResNet-28-2) and if they do, why is fine-tuning still increasing the accuracy?
- Minor: Try to use consistent coloring at Figure 6 as while (a), (b) share the same color-coding, (c) is using a different one hence could be confusing.

[1] Estimating or Propagating Gradients Through Stochastic Neurons for Conditional Computation, Yoshua Bengio, Nicholas Léonard, Aaron Courville, 2013
[2] Understanding Straight-Through Estimator in Training Activation Quantized Neural Nets, Penghang Yin, Jiancheng Lyu, Shuai Zhang, Stanley Osher, Yingyong Qi, Jack Xin, 2019

**Experience Assessment:**

I have published one or two papers in this area.

**Review Assessment: Checking Correctness Of Derivations And Theory:**

I assessed the sensibility of the derivations and theory.

**Review Assessment: Checking Correctness Of Experiments:**

I assessed the sensibility of the experiments.

**Review Assessment: Thoroughness In Paper Reading:**

I read the paper thoroughly.

---

> ### Author Response · Authors · 2019-11-12
> **Response to Reviewer1**
>
> Thank you for your review.
>
> We have updated the draft to address your concerns.
> In particular, we explain the connection to the error feedback framework better (see footnote on page 4 which makes the connection very explicit), fixed the typo, updated the colors in Figure 6 and added a short discussion of the relation to STE.
>
> We did further clarify why we think that DPF can profit from fine-tuning: Whist Figure 4 shows that a large fraction of the mask elements converge, however, a few elements are still fluctuating even at the end of training (approximately 5% (depending on dataset/model) of the active (non-pruned) weights). Thus, after fixing the final mask, fine-tuning of the weights to the chosen mask can provide additional benefits. A similar behavior (Figure 7) can be found for ResNet-20 on CIFAR-10.

---

> > ### Comment · AnonReviewer1 · 2019-11-14
> > **Response to rebuttal**
> >
> > Thank you for addressing my comments. There is a minor typo in the revised version page 7, section 6, second sentence "grantees" -> "guarantees".
> >
> > I will leave the score unchanged and vote for accepting this work.

---

### Official Review · AnonReviewer3 · 2019-10-23
**Official Blind Review #3**

**Rating:** 6

**Review:**

In this paper, the authors proposed a novel model compression method that uses error feedbacks to dynamically allocates sparsity patterns during training. The authors provided a systematic overview of a good number of existing model compression algorithms depending on the relative order of pruning and training processes. The effectiveness of the proposed algorithm is illustrated by comparing its generalization performance with 6 existing algorithms (and their variants) with two standard datasets and various networks of standard structures. The authors also showed the convergence rate and the fundamental limit of the proposed algorithm with two theorems.

This paper is well-written and very pleasant to read. I would like to accept this paper. But since I have never actually done research in model compression, I would say this is my 'educated guess'.

Some quick comments:
1. I did not go through the proofs of the two theorems. But it seems that there is a typo in the definition of strong convexity on Page 4: '\Delta f(w)' should be '\Delta f(v)'. I assume that this is just a typo.
2. Sorry again for not knowing the details of the baseline algorithms. According to Table 1 and Table 2, the proposed method (DPF) outperforms all the baseline algorithms, without a single exception, which looks suspicious for me. After reading the paper, I still don't understand why this should be the case. Is this due to some implementation details? Can you think of some scenarios that the proposed algorithm may not be the one to go with? In other words, when the experiment seems to show that one algorithm absolutely outperforms all the other existing algorithms, there should be some take-home message on why, or some known limitations of the proposed method.


**Experience Assessment:**

I do not know much about this area.

**Review Assessment: Checking Correctness Of Derivations And Theory:**

I did not assess the derivations or theory.

**Review Assessment: Checking Correctness Of Experiments:**

I assessed the sensibility of the experiments.

**Review Assessment: Thoroughness In Paper Reading:**

I made a quick assessment of this paper.

---

> ### Author Response · Authors · 2019-11-12
> **Response to Reviewer3**
>
> Thank you for your review. We have fixed the typo (1.) in our revision. We hope we can clarify your concerns (2.) on the performance of DPF:
>
> [Superior performance]
> The superior performance of our method originates in the flexible error feedback scheme.
> Our scheme can be incorporated with different pruning criteria with less hyper-parameter tuning than other, more specialized, approaches. We believe that the generality and simplicity of our scheme enable good performance across all tasks. Consequently, we expect algorithms that are fine-tuned to specific architectures or tasks could perform better, though we did not observe this in the experiments so far.
>
> The superior performance is not due to the implementation details. All our evaluations are performed under a fair experimental setup by using a similar pruning configuration, the released codes and recommended hyper-parameters for the competitor methods. A side note for Table 2: for pruning model with limited capacity in dense space (e.g. ResNet-20) and high target sparsity ratio (e.g. 95%), our method sometimes cannot find a much better sparse model than Incremental (ZG, 2017) if no fine-tuning is involved.

---

### Official Review · AnonReviewer2 · 2019-10-23
**Official Blind Review #2317**

**Rating:** 6

**Review:**


Main contribution of the paper
- The paper proposes a new pruning method that dynamically updates the sparse mask and the network weight.
- Different from the other works, the proposed method does not require post-tuning.
- A theoretical explanation of the method is provided.

Methods
- In this method, the weight of the baseline network is updated not by the gradient from the original weight but pruned weight.
- Here, pruning can be conducted by (arbitrary) a pruning technique given the network weight (Here, the author uses the magnitude-of-the-weight given method from Han.et.al).


Questions
- See the Concerns

Strongpoints
- The author provides the simple and effective pruning method and verifies the performance with a sufficient amount of experiments.
- The author argues that the method is applicable to various pruning techniques.

Concerns
- It seems that the paper omits the existing work (You.et.al - https://arxiv.org/pdf/1909.08174.pdf), which seems to share some contribution. The reviewer wants the author to clarify the differences and the strongpoints compared to the work.
- The main pruning&update equation (DPF) does not seem to force the original network w to become sparse, such as by l1-regularization. So, the reviewer worried that the method might not produce sparsity if the initial weights are not that sparse.
If the reviewer missed the explanation about this, clarify this.
- Regarding the above concern, what if we add regularization term in training the original network w?
- As far as the reviewer knows, the proposed method improves the sparsity of the network, but most works choosing the strategy actually cannot meaningfuly enhance the operation time and just enhances the sparsity. Does the author think that the proposed method can enhance the latency? If so, a detailed explanation or experiment will be required.

Conclusion
- The author proposes a simple but effective dynamic pruning method.
- The reviewer has some concerns regarding the novelty, real speed up, and guarantee of the sparsity.
However, the reviewer thinks that this work has meaningful observations for this field with a sufficient amount of verification, assuming that the author's answers for the concerns do not have much problem.

Inquiries
- See the Concerns parts.

**Experience Assessment:**

I have read many papers in this area.

**Review Assessment: Checking Correctness Of Derivations And Theory:**

I assessed the sensibility of the derivations and theory.

**Review Assessment: Checking Correctness Of Experiments:**

I carefully checked the experiments.

**Review Assessment: Thoroughness In Paper Reading:**

I read the paper thoroughly.

---

> ### Author Response · Authors · 2019-11-12
> **Response to Reviewer2**
>
> Thank you for your review. We have updated the draft and answer below to your specific questions:
>
> [Connection to You et al]
> Thank you for pointing out the recent parallel work You et al. We have cited this work in the related work section. We explain the key differences below:
> 1. The Tick-Tock framework introduced in You et al. is only validated on filter pruning while our current submission focuses on unstructured weight pruning (mainly) and filter pruning.
> 2. The Tick-Tock pruning framework (Figure 2 in You et al.) requires a pre-trained model while our method allows to (1) train a compressed model from scratch with trivial extra cost, and (2) pruning a pre-trained model (we will add additional new experimental results confirming this application to the appendix).
> 3. Our method is simpler and easier to implement than Tick-Tock. Tick-Tock involves multiple phases of pruning and finetuning: the Tick phase learns the filter importance with the subset of data samples, and the Tock phase fine-tunes the sparse model on the full data samples. Instead, our method reparametrizes the sparse model via a standard single training pass.
> 4. The Tick-Tock framework is more close to ZG17 than to DPF. They finetune/tock the sparse model while we update the model on the dense space via the error-feedback scheme.
>
> [Without ‘forcing’ the sparsity]
> We agree with the reviewer that our method does not use l1-regularization to `force` the original weight w to be sparse. Instead, we directly prune weights by increasing order of importance, until reaching our specified target sparsity (magnitude-based pruning) and our error feedback training scheme allows the weight to be flipped back to recover the damage from improper pruning. Even though the initial weights are not sparse, our method will always reach the expected target sparsity (w.r.t. the considered layers) after training.
>
> In comparison to L1-based methods, our approach has no additional pruning-specific hyperparameters, and thus simplifies usage while still reaching SOTA results. We directly use the hyperparameters from the original training scheme of the dense model.
>
> [The training efficiency]
> As demonstrated in Figure 12 in the Appendix, our proposed method enables to train a sparse model from scratch with trivial computational overhead for task on the scale of Imagenet.
>
> The current submission focuses on verifying the effectiveness of the proposed method (in terms of test performance). A more efficient implementation can further improve the training efficiency for better speedup, for instance, (1) get the gradients at the sparse model (mentioned in the footnote at page 3), (2) automatically control the reparameterization space by using the runtime information (e.g. as shown in Figure 4). We leave such specific improvements for future work.

---

### Public Comment · ~Utku_Evci1 · 2019-10-02
**Connections to Model Pruning**

Hi,

I just read your paper. It is well written and has comprehensive experiments. I'd liked to ask a few questions:

1) Looking at your method, I am not sure I understand the difference between your method and Zhu, 2017 (ZG17)'s. At related work it says: '...where pruned weights are not allowed to flip back.' I am afraid this is not true. Their code can be found here: https://github.com/tensorflow/tensorflow/blob/r1.14/tensorflow/contrib/model_pruning/python/pruning.py . They change the mask without setting the pruned connections to 0, so that they can become alive again in the next pruning iteration. Looking at their code your algorithm and theirs seem quite similar, except the global pruning criteria.  I might be missing something. Would you mind clarifying the differences between your work and ZG17's?

2) I think you should compare your results with Gale, 2019 (https://arxiv.org/abs/1902.09574) since they use ZG17's code and do proper hyper-param tuning. Their results, as far as I know, SOTA for Resnet-50 pruning and it is the same exact method as ZG17. Their results can be found here: https://github.com/google-research/google-research/tree/master/state_of_sparsity .

Thank you

---

> ### Author Response · Authors · 2019-10-04
> **Response to "Connections to Model Pruning"**
>
> Thanks for your interest! Our method is different from ZG17 in three main points illustrated below, and we did compare with (and outperform) fine-tuned ZG17 (our ZG17 baseline has similar performance as the one in Gale19 under a fair comparison).
>
> [Answer to Q1: Key differences between ZG17 and our scheme]
> (i) ZG17 does not update weights in the dense model that are currently masked/pruned; in our scheme we apply the gradient updates to *all* weights in the dense model.
> (ii) ZG17 updates the mask only when the sparsity is changed (according to a prescribed schedule) while our scheme updates the mask periodically (independent of the current sparsity ratio). An ablation study of the reparameterization period is illustrated in Figure 9a (page 16) in our Appendix. In both schemes, pruned weights can ‘flip back’. We will update this inaccurate statement in the next revision.
> (iii) In ZG17, once the model achieves the target sparsity, the weight masks are no longer updated. In contrast, we perform dynamic reparameterization (i.e. changing the mask) over the whole training procedure, even when the target sparsity is reached. An illustration of the mask flipping behavior during the training can be found in Figure 4 (page 8).
>
> For comparison, we derived a similar plot for ZG17 that we will include in the next revision. The data shows that our scheme changes up to extra ~40% more of the mask elements than ZG17, and thus explores a larger space.
>
> [Answer to Q2: Comparing to fine-tuned ZG17 implementation in Gale19]
> We did compare to ZG17. Our implementation of ZG17 had a similar quality drop (for ResNet50 on ImageNet) as in Gale19 (under a fair comparison). The detailed explanations are below:
> (1) In Gale19, they trained ResNet50 on ImageNet by increasing the number of training steps (1.5x), in terms of extending the region when performing the gradual pruning scheme. Thus, the total number of training epochs (to achieve the best performance) in Gale19 is increased from (the standard one) 90 epochs to 105 epochs. The increased training epochs/flops are significant in terms of ImageNet scale experiments.
> (2) For ImageNet experiments (and other experiments in our paper), we focused on performing a fair comparison (in Table 3 on page 7) where every method uses the same and standard training scheme (e.g. the number of epochs, learning rate schedule) and thus in the paper we only compared with ZG17 (we fixed the gradual pruning scheme). We also carefully checked the results in Gale19 and below are the differences (by default all methods use the same training epochs):
> * Top-1 acc drop (0.80 target model sparsity)
>     * Our reimplementation of ZG17: -1.70
>     * Gale 19 implementation of ZG17: -1.10
>     * Our scheme DPF: -0.47
> * Top-1 acc drop (0.90 target model sparsity)
>     * Our reimplementation of ZG17: -2.59
>     * Gale 19 implementation of ZG17: -2.80
>     * Gale 19 implementation of ZG17 + extra 15 epochs: -1.60
>     * Our scheme DPF: -1.44
> Note that we picked the best performance from [1] for each sparsity level and calculated the quality loss compared to the baseline performance. Also, note that unlike Gale19, all methods evaluated in our paper DO NOT use label smoothing, which is known as a powerful trick to (potentially) improve the performance of ImageNet training.
> (3) We would like to emphasize that adding more training tricks (e.g. label smoothing, mixup) to improve the performance is orthogonal to our work, as it can improve the performance for both dense baselines and pruned models. Also, as pointed out by Table 2 (page 7), DFP+finetuning (FT) can further (significantly) improve the performance, which is much better than ZG17 + FT.
>
> ----
> References
> [1] https://github.com/google-research/google-research/blob/master/state_of_sparsity/results/sparse_rn50/technique_comparison/rn50_magnitude_pruning.csv

---

> > ### Public Comment · ~Utku_Evci1 · 2019-10-10
> > **Additional Comments and Questions**
> >
> > Thanks for your reply, it helps clear some things up, but I still have some questions.
> >
> > 1) Similarities to the Model Pruning:
> > (i) I missed the dense gradient update part. That is indeed different than the model pruning. Thanks for the clarification.
> > (ii) There are few other places in the paper where it is implied that the current pruning algorithms are unable to recover from their mistakes, i.e. Page-4: 'in contrast to incremental pruning approaches that have to stick to sub-optimal decisions' and Figure-1: 'Cannot recover from premature pruning.' It might be accurate to update those, too.
> > (iii) With ZG17's library, It is possible to update connections after the target sparsity is reached using 'end_pruning_step' and 'sparsity_function_end_step' arguments. Default value for end_pruning_step seems to be -1, which enables mask updates until the end of the training.
> >
> > 2) Model Pruning Baseline
> > I agree comparing baselines across different settings is a bit tricky. The comparison of models with 73.5% (yours) and 80% sparsity (Gale19), and then also 82.6% sparsity (yours) and 90% sparsity (Gale19) seems inaccurate.
> > Given the numbers available to us, it seems like the best comparison we can make is between models with 80% sparsity (Gale19) and 82.6% sparsity (yours).
> > In this case, I note that Gale19 achieves -1.10 accuracy drop while your technique achieves -1.44.  When allowing for longer training Gale19 achieves -0.16 accuracy drop.
> >
> > (3) Grouping of Methods
> > Dynamic-incremental pruning separation is not clear to me. Is the main difference that the sparsity of the model oscillates around a level (mid, low, high) (see Fig:1)? It is also not clear to me why the sparsity oscillates during training for DPF.
> > Both ZG17 and your proposed method require training to be dense, while the methods of SM and DSR do not, perhaps they should be categorized differently. They are more close to the 'prune before training' methods than to the 'dynamic' ones.
> >
> > (4) Uniform vs Non-Uniform Layer Sparsities
> > Gale19 and ZG17 uses uniform sparsity across layers which makes total FLOPs to scale with (1-sparsity) directly. However your method employs global pruning and allows redistribution of sparsity across layers. We know that global methods tend to sparsify earlier layers less aggressively [Liu17] increasing total FLOPs. It would be nice to see how resulting sparsity distribution looks like with DPF and its effect to the total FLOPs.

---

> > > ### Author Response · Authors · 2019-10-16
> > > **Response to "Additional Comments and Questions"**
> > >
> > > Thank you for your questions!
> > >
> > > [Answer to Q1 and Q3: Understanding and Grouping of Methods]
> > > Both ZG17 and Gale19 terminate the mask update when reaching the target sparsity. The ability to update the mask after reaching the target sparsity is implemented, but not used in ZG17 (up to our understanding). Also, Gale19 does not use this feature (the same Tensorflow API and do intensive hyper-parameters tuning; [1]). Our previous statements are on top of these observations and thus argue that ZG17 might not recover from premature pruning.
> > >
> > > As mentioned in our previous answer, we will carefully polish the related work section and the grouping of the methods as by your suggestion. However, please note that the statement in Figure 1 only applies to 'incremental methods' that can by definition (see also Figure 1) not recover from pruning errors.
> > >
> > > [Answer to Q2: Model Pruning Baseline]
> > > We performed a fair comparison with our baselines (by using their codes and recommended hyper-parameters) in our current submission; we reached similar sparsity (same sparsity evaluation function w.r.t. whole model) under the same target sparsity, where we ignored the pruning for the bias, bn, and fully connected layers as most of the papers. Note that most papers only report the reached sparsity for the pruned layers while our reported ‘reached sparsity’ is in terms of the whole model.
> > >
> > > We agree it is difficult to directly compare our previous ImageNet results with Gale19 due to different sparsity ratios and different type prune layers. Thus, we present our new results below (only ignore bias and bn layers as in Gale19), where we train for the same 90 epochs and reach the same sparsity (w.r.t. the whole model).
> > >  * top-1 acc drop (0.80 target model sparsity)
> > >       * DPF: -0.82
> > >       * Gale19: -1.10
> > >
> > > [Answer to Q4: Uniform vs Non-Uniform Layer Sparsities]
> > > This is an interesting question. We will try to provide an ablation study about how the sparsity ratio over layers (uniform/non-uniform layer sparsities) will impact the total FLOPs. Also, w.r.t. your comment about Gale19 the results provided by Gale19 also use non-uniform sparsities across layers [2].
> > >
> > > -----
> > > Reference.
> > > [1] Looking at the code from line 276 to line 289 at  https://github.com/google-research/google-research/blob/master/state_of_sparsity/sparse_rn50/imagenet_train_eval.py, we can witness that the used `end_pruning_step` is equivalent to `sparsity_function_end_step`, indicating that the mask will not be updated after reaching the target sparsity.
> > > [2] https://github.com/google-research/google-research/tree/master/state_of_sparsity#trained-checkpoints

---

> > > > ### Public Comment · ~Erich_K_Elsen1 · 2019-10-21
> > > > **Some Thoughts**
> > > >
> > > > For what it's worth, I've preferred to stop updating the mask when the final sparsity is reached because it combines well with EMA, which is useful in some problems (like RNNs for TTS).   It also (anecdotally) hasn't seemed to help to let it continue updating.
> > > >
> > > > As someone who would very much like a better pruning technique for training models to deploy it would be very helpful to understand if the increased performance is due to increasing the FLOPs required for inference.  If the accuracy can be increased under the same FLOP budget that would be more useful (to me anyway) in practice than a constant parameter budget.  I look forward to the ablation.
> > > >
> > > > For the models that you are comparing with (80% sparse, original or extended training), the only non-uniformity in the models from Gale19 is that the first layer is dense and all remaining layers have a uniform sparsity.  For the higher sparsity extended training models the final fully connected layer remains at 80% and the sparsity of the intermediate layers is adjusted higher to compensate.  All intermediate layers still have the same sparsity.

---

### Author Response · Authors · 2019-11-12
**Revision 1**

We would like to thank all reviewers for their valuable comments and questions. We have prepared an updated version of the manuscript (fixing typos mentioned by the reviewers and added few clarifications as mentioned in the other comments).

---

### Public Comment · ~Shangyu_Chen1 · 2019-12-30
**Connection with Dynamic Network Surgery**

Hi !

Congratulation on your acceptance!
I got a small question when reading your paper: I found the proposed algorithm (if I understand correctly) is very similar to Dynamic Network Surgery (NIPS2016), as: 1) Both of the papers update the mask during training (Eq.DPF is similarly used in both papers). 2) Full-precision weights are updated even pruned, so that pruned weights can be recovered. 3) Both of the papers use iterative (dynamic) pruning to incrementally increase sparsity.

I notice you also cite Dynamic Network Surgery, but is is in "Pruning after training". In my understanding,  Dynamic Network Surgery should be classified into "Pruning during training" since it continuously prunes according to the update of network (thus similar with the framework employed in this paper).

I may make some misunderstanding in several points. Thanks in advance if you can correct and solve my concerns.

Best regards,
Shangyu

---

> ### Author Response · Authors · 2020-01-16
> **Clarify**
>
> Thank you for your interest in our paper.
>
> We would like to point out that Dynamic Network Surgery (DNS) prunes after the training while our DPF performs dynamic reparameterization during the training. More precisely, DNS requires to first train a dense model from scratch through a standard training procedure. Its pruning is only performed on a well-trained model. This pattern is illustrated in Figure 2 and Table 1 of their paper, and thus DNS should be classified into the category of "pruning after the training".
>
> The general way of updating the model is similar in DNS and DPF. However, it differs in the following points:
> * DPF generalizes the idea of DNS and provides a general convergence analysis. Our improved solution (DPF) can achieve SOTA model compression results even without extra fine-tuning.
> * We simplify the masking function and avoid introducing two extra hyper-parameters (their equation 3) as in DNS.
> * Our training is end-to-end. DNS requires to prune the convolutional layers and fully connected layers separately to avoid vanishing gradient problem.
> * DNS reparameterizes the model around the local minima (pruning after the training) while DPF dramatically reparameterizes the model during the training (from scratch). The reparameterization differences in different phases are illustrated in our paper (Figure 4 and Figure 7).
> * The intuition of dynamic reparameterization in DNS and DPF is different:
>     * The mask update scheme in DNS is triggered stochastically. For convergence reasons, the triggering probability is monotonically non-increasing and towards 0 in terms of update steps. BTW, up to my understanding, the pruning sparsity in DNS is not incrementally increased.
>     * DPF uses a constant reparameterization step over the whole training procedure where the mask will automatically converge to a stable state. We avoid the careful triggering function design used in DNS.

---

### Decision · Program_Chairs · 2019-12-19

**Decision:**

Accept (Poster)

**Comment:**

The paper proposes a new, simple method for sparsifying deep neural networks.
It use as temporary, pruned model to improve pruning masks via SGD, and eventually
applying the SGD steps to the dense model.
The paper is well written and shows SOTA results compared to prior work.

The authors unanimously recommend to accept this work, based on simplicity of
the proposed method and experimental results.

I recommend to accept this paper, it seems to make a simple, yet effective
contribution to compressing large-scale models.